# Bioinspired one-pot furan-thiol-amine multicomponent reaction for making heterocycles and its applications

Yuwen Wang [1], Patrick Czabala [1] & Monika Raj [1] ✉

One-pot multicomponent coupling of different units in a chemoselective manner and their late-stage diversification has wide applicability in varying chemistry fields. Here, we report a simple multicomponent reaction inspired by enzymes that combines thiol and amine nucleophiles in one pot via a furan-based electrophile to generate stable pyrrole heterocycles independent of the diverse functionalities on furans, thiols and amines under physiological conditions. The resulting pyrrole provides a reactive handle to introduce diverse payloads. We demonstrate the application of Furan-Thiol-Amine (FuTine) reaction for the selective and irreversible labeling of peptides, synthesis of macrocyclic and stapled peptides, selective modification of twelve different proteins with varying payloads, homogeneous engineering of proteins, homogeneous stapling of proteins, dual modification of proteins with different fluorophores using the same chemistry and labeling of lysine and cysteine in a complex human proteome.

The term orthogonal refers to the 100% selective coupling reactions between different complementary pairs in the presence of reactive functional groups. Orthogonal reactions include CuAAC, oxime, hydrazone, thiol-ene and Diels Alder cycloaddition reactions[1–7]. These orthogonal reactions are widely used for applications ranging from synthesis of small molecules, biomolecules, polymers, gels and materials to late-stage diversification of natural products, bioactive compounds, peptides, antibody-drug conjugates and cell imaging[8–12]. Most of these orthogonal reactions involve two reactive components although there are few exceptions such as Ugi (4-MCR), Passerini (3-MCR), Mannich (3-MCR) and Petasis (3-MCR) reactions that are applicable to biomolecules[13–17]. Our group recently utilized the Petasis reaction for the selective labeling of proteins with N-terminal proline[18]. Inspired by the well-known cytochrome P450-catalyzed oxidation of furan to *cis*−2-butene-1,4-dial (BDA), which reacts with glutathione (GSH) and cellular amines to generate thio-*N*-pyrroles cross-linked products[19–23], we re-envisioned this observation as a highly selective, multicomponent reaction (3-MCR) (Fig. 1). Previously, this reaction has been extensively explored for determining the biomarkers associated with furan toxicity by analyzing the cross-linked proteins due to the low volatility of the furan[19–23]. Recently, Zheng et al. explored this method for chemoproteomic profiling of lysine and cysteine on proteins in a complex cellular mixture[24]. Inspired by these observations, we sought to develop this into a one-pot orthogonal MCR, termed Furan-Thiol-Amine (FuTine), that selectively couples electrophilic BDA with nucleophilic thiols and amines to generate *N*-pyrrole heterocycles. The unique feature of this chemistry is that all three components couple in sequential order where BDA first reacts with a thiol followed by a reaction with an amine to generate an exclusive *N*-pyrrole product in high yields.

Here, we showcase that the FuTine reaction is orthogonal and proceeds independently of the substitutions on furan, thiol and amine under physiological conditions, thus making it ideal for selective modification of biomolecules (Fig. 1). This robust and highly selective orthogonal 3-MCR expands the breadth of functional properties that can be introduced in small molecules, polymers, peptides, proteins and cells that cannot be accessed through traditional 2-MCRs. We show the application of FuTine chemistry in late-stage diversification of peptides, synthesis of macrocyclic and stapled peptides, precision engineering of proteins, homogeneous stapling of proteins, dual labeling of proteins with varying payloads using the same chemistry and selective fluorophore labeling of proteins in a complex cell lysate.

[1]Department of Chemistry, Emory University, 30322 Atlanta, GA, USA. ✉e-mail: monika.raj@emory.edu

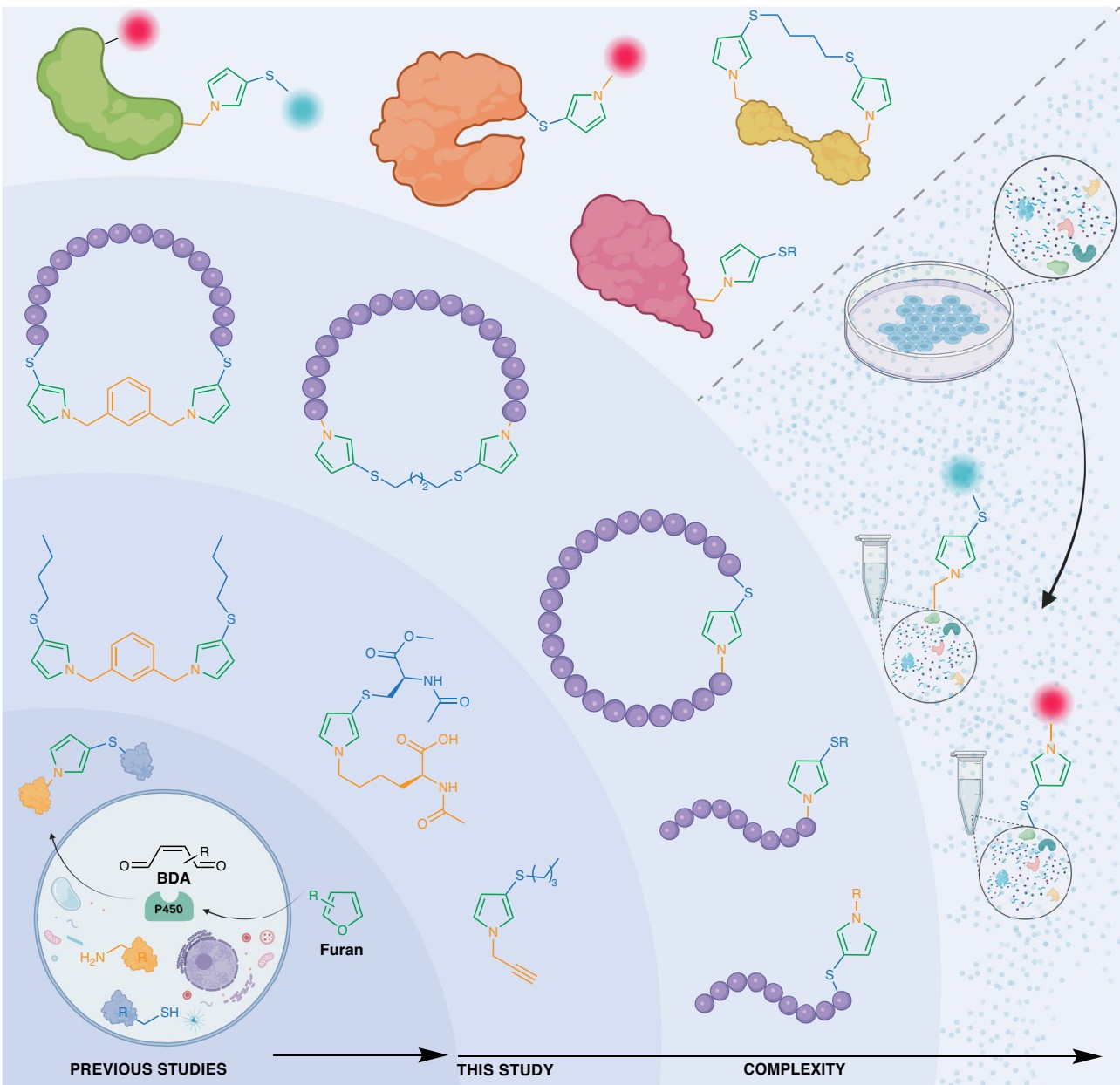

**Fig. 1 | Bioinspired one-pot furan-thiol-amine (FuTine) multicomponent reaction (MCR).** Previous studies: The identification of the toxicity of furan by analyzing the protein crosslinking products obtained by the reaction of amine and thiols on proteins with electrophilic *cis*−2-butene-1,4-dial (BDA) generated by oxidation of furan by enzyme P450. This study: Application of FuTine chemistry in the synthesis of varying small *N*-pyrroles, selective labeling of peptides and proteins, macrocyclization and stapling of peptides, precise protein engineering, homogeneous stapling of proteins, dual modification of proteins and selective labeling of lysine and cysteine in whole proteomes in a complex cell lysate mixture. Created with BioRender.com.

This research further expands the repertoire of synthesis and biological labeling strategies, creating new possibilities at the interface of chemistry and biology.

## Results

### Design and optimization of one-pot FuTine multicomponent reaction

Inspired by the rapid conversion of furan to *N*-pyrrole inside the cell under cytochrome P450-catalyzed conditions[19–24], we reasoned that the selective oxidation of furan and sequential reaction with thiols and amines might serve as an attractive starting point for the synthesis of *N*-pyrroles and selective labeling of biomolecules with diverse functionalities. This is largely due to the high reactivity and efficiency of this MCR and its tolerance of substituents on all three components. We

initiated our investigation with the oxidation of furan **1** using *N*-bromosuccinimide (NBS) in $CH_3COCH_3:H_2O$ as the solvent to generate *cis*−2-butene-1,4-dial (BDA) in situ followed by a sequential reaction with 1-butanethiol and 1-butylamine in one pot to generate a single *N*-pyrrole heterocyclic product **2a**. In our early studies, 1-butanethiol was introduced first to the BDA intermediate and allowed to react for 30 min before the addition of 1-butylamine (entries 1–2, Fig. 2a, Supplementary Fig. 1, Procedure A and B). Subsequently, we found that the thiol and amine can be added simultaneously to the BDA intermediate to still generate the same 3-thio *N*-pyrrole heterocycle **2a** because of the high nucleophilicity of thiols as compared to amines (entry 3, Fig. 2a, Supplementary Fig. 1, Procedure C). We further optimized the reaction conditions for equivalents of reagents, sequential addition and temperature, resulting in the formation of *N*-pyrrole heterocycle

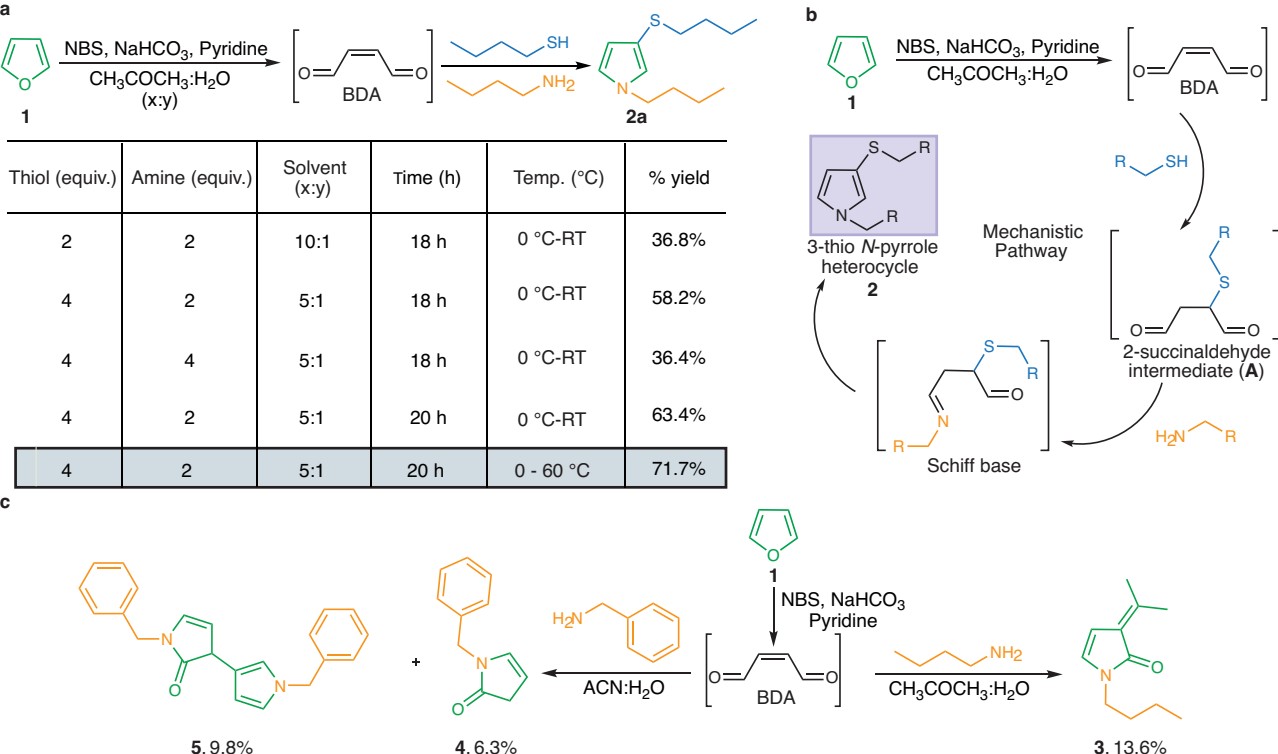

**Fig. 2 | Development of one-pot furan-thiol-amine (FuTine) multicomponent reaction (MCR). a** Optimization of furan-thiol-amine (FuTine) reaction for the formation of *N*-pyrrole products in one-pot by varying equivalents of reagents, temperature and the ratio of solvents. **b** A proposed mechanistic pathway involving the oxidation of furan to generate *cis*−2-butene-1,4-dial (BDA) for reaction with thiol by 1,4-addition followed by trapping with amine, rearrangement and aromatization to generate an *N*-pyrrole heterocycle. **c** The formation of small amounts of side products in the absence of thiols by direct reaction of oxidized furan (BDA) with amines. In acetone, 3-alkylidene-4-*N*-butylpyrrolin-2-one **3** was obtained in poor yield. In acetonitrile, 2-pyrrolone **4** and unique dimer **5** were obtained with poor yields.

**2a** in good yields (63.4–71.7%) (entries 4 and 5, Fig. 2a, Supplementary Fig. 1, Procedure D). The unique selectivity of BDA for coupling a thiol and an amine arises from its structure. Thiols are more nucleophilic than amines and thus the thiol reacts first with BDA via 1,4-addition to generate a 2-succinaldehyde intermediate (**A**) (Fig. 2b). Intermediate **A** further gets trapped by an amine to form a Schiff base followed by rearrangement and aromatization to generate a 3-thio *N*-pyrrole heterocycle **2**. Thiols are soft nucleophiles and thus do not add twice to 2-succinaldehyde intermediate **A**. We did not observe the formation of any stable product in the absence of amines. In contrast, amines can react directly with BDA in the absence of thiol to form multiple products such as 2-pyrrolones or 2-pyrrolidinone, though in poor yields[25,26]. Surprisingly, the reaction of BDA with butylamine alone generated 3-alkylidene-4-*N*-butylpyrrolin-2-one **3** via the trapping of acetone by 2-*N*-butylpyrrolone in poor yield (13.6%). The resulting 3-alkylidene-4-*N*-butylpyrrolin-2-one **3** product was characterized by nuclear magnetic resonance (NMR) and high-resolution mass spectrometry (HRMS) (Fig. 2c and Supplementary Fig. 2).

In the absence of acetone, we observed the formation of 2-pyrrolone **4** (6.3%) along with the formation of a unique dimer **5** (9.8%) obtained by the reaction of 2-pyrrolone with furan coupled with amine (Fig. 2c and Supplementary Fig. 3). The products **4** and **5** were characterized by NMR and HRMS (Supplementary Fig. 3). In sum, the reaction of BDA with amines in the absence of thiol is poor yielding, and we observed complete suppression of this side product in the presence of thiol.

For further studies, we focused our attention on using the FuTine MCR to generate *N*-pyrrole heterocycles under aqueous conditions. With the optimized conditions in hand, we determined the scope of the FuTine MCR by carrying out reactions with various pairwise combinations of thiols and amines containing different reactive functional groups such as acids, alcohols, alkynes, amides and esters (Fig. 3a and Supplementary Fig. 4). In all cases, high yields of 3-thio *N*-pyrrole products **2** were obtained (**2b-2i**, 50–75%) independent of the nature of thiols and amines. Interestingly, the reaction with 2-aminoethanol generated 3-thio *N*-pyrrole **2h** (52.3%) as a major product along with the formation of a small amount of 2-thio *N*-pyrrole **2h'** (14.7%) as confirmed by the NMR (Supplementary Fig. 4). We hypothesized that it might be due to the reactivity of the hydroxyl group of 2-aminoethanol that forms oxazolidine with an aldehyde followed by the nucleophilic attack of thiol at the reactive oxazolidine intermediate to generate 2-thio *N*-pyrrole as a minor product (Supplementary Fig. 4). Similarly, a small amount of 2-thio *N*-pyrrole was observed on reaction with glycine generating **2f'** (5.3%) along with the major product 3-thio *N*-pyrrole **2f** (53.9%) as analyzed by the NMR of the inseparable mixture of **2f** and **2f'** (Fig. 3a and Supplementary Fig. 4). A similar observation for the formation of minor 2-thio *N*-pyrrole has also been reported previously[27]. Notably, less nucleophilic aromatic amines such as aniline also reacted with oxidized furan and 1-butanethiol to generate 3-thio *N*-pyrrole **2j** in high yields (69.7%). The reaction was further applied for forming pyrrole products with varying amino acids such as *N*-acetyl cysteine, phenylalanine methyl ester and *N*-acetyl lysine to generate 3-thio *N*-pyrrole heterocycles (**2e, 2i,** and **2k,** respectively) in high yields (55.1-74.6%), confirming the broad scope of FuTine MCR in selective modification of amino acids (Fig. 3a and Supplementary Fig. 4).

The reaction of the substituted 2-methyl furan with 1-butanethiol and 1-butylamine also generated 3-thio *N*-pyrrole product **2l** in good yield (62.4%, Fig. 3a and Supplementary Fig. 4). Next, we explored the FuTine MCR with diamines such as 1,3-bis(aminomethyl)benzene and

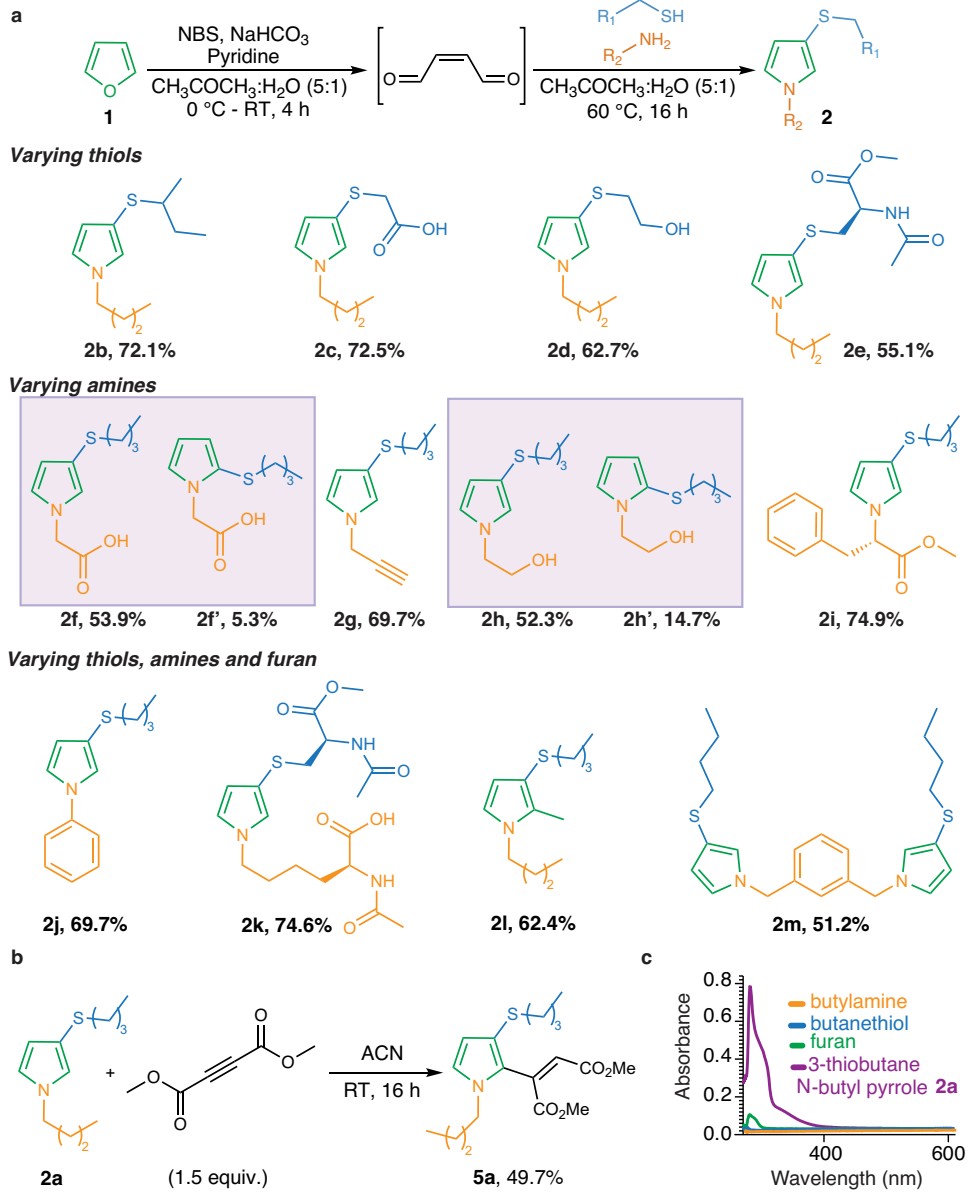

**Fig. 3 | Substrate scope of furan-thiol-amine (FuTine) multicomponent reaction (MCR) for the synthesis of small *N*-pyrrole molecules. a** Substrate scope with varying furan and functionalized thiols and amines. The reaction is compatible with varying functional groups such as acids, alcohols, alkynes, amides and esters. The reaction generated good yields of *N*-pyrrole products with aniline, various amino acids and 1,3-bis(aminomethyl)benzene. **b** Late-stage modification of *N*-pyrrole by reaction with dimethyl acetylenedicarboxylate (DMAD). **c** The aromatic *N*-pyrrole product showed characteristic absorbance at 280 nm in contrast to the starting materials of this MCR (furan, thiol and amine), which show little or no absorbance.

observed the dual modification of both the amines to 3-thio *N*-pyrroles (**2m**, 51.2%) on reaction with oxidized furan and 1-butanethiol (Fig. 3a and Supplementary Fig. 4). This reactivity is distinctly different from that of other homobifunctional crosslinkers of thiols and amines, which are dependent on specific reaction conditions and the identity of thiol and amine[28,29]. Overall, this study showed that the reaction is highly robust and generates 3-thio *N*-pyrrole heterocycles independent of the nature of furan, thiol and amine, thus demonstrating its potential applicability to multiple areas of chemistry.

**Late-stage functionalization of 3-thio *N*-pyrroles**
It is challenging to couple three components in one pot to generate one stable product selectively. Furthermore, the late-stage modification of the multicomponent product to introduce new functionalities is an unsolved challenge. To explore the possibility of modification of

the *N*-pyrrole ring, we incubated 3-thiobutane *N*-butylpyrrole **2a** with dimethyl acetylenedicarboxylate (DMAD) for 16 h at room temperature (Fig. 3b and Supplementary Fig. 5). The reaction resulted in the functionalization of the pyrrole ring at the second position with an alkene handle to give **5a** as confirmed by $^1$H-NMR and $^{13}$C-NMR (Fig. 3b and Supplementary Fig. 5). We hypothesized that the *N*-pyrrole ring acts as a nucleophile and adds to the reactive acetylenedicarboxylate leading to the functionalization by one unit of acetylenedicarboxylate at 2nd position only. This post-modification of the pyrrole ring further diversifies the available structural complexity of the conjugates.

**Photophysical properties of 3-thio *N*-pyrroles**
Since the FuTine MCR generates an *N*-pyrrole heterocyclic ring, we sought to explore its photophysical properties by using a UV spectrophotometer. We begin our studies with 3-thiobutane *N*-butylpyrrole

**2a**, which showed a characteristic absorbance for the pyrrole ring at 280 nm (Fig. 3 and Supplementary Fig. 6). In contrast, starting components furan, 1-butanethiol and 1-butylamine did not show significant absorbance in this region. This characteristic absorbance of the *N*-pyrrole conjugate was useful for multiple applications including monitoring the reaction progress by TLC (Thin Layer Chromatography) and HPLC (High-Performance Liquid Chromatography) and purification of the conjugate product by HPLC and silica-gel column chromatography.

## Late-stage diversification of peptides by FuTine MCR

The broad scope with small molecules showed that the reaction is chemoselective between furan, thiol and amine in the presence of other reactive functional groups. Moreover, the reaction works under mild aqueous conditions thus exhibiting potential applicability to varying fields of chemistry including modification of peptides. In this regard, we investigated the lysine specificity of FuTine MCR by carrying out the modification of a peptide Ac-KNSRY **1n**, containing reactive amino acids Tyr, Arg, Ser and Asn along with lysine at the *N*-terminus using *N*-acetyl cysteine ester as a thiol source in the presence of oxidized furan in ACN:H$_2$O (5:1) (Fig. 4a). The reaction afforded 87.9% of *N*-pyrrole peptide product **2n** with lysine within 4 h at room temperature (Supplementary Fig. 7).

Next, we selectively modified the side chain of Cys in a peptide Ac-CMHWD **1o** containing the reactive amino acids Met, His, Trp and Asp using propargylamine as the amine source in the presence of oxidized furan to generate the *N*-pyrrole peptide product **2o** in 78.7% yield thus demonstrating high specificity and reactivity towards cysteine (Fig. 4a and Supplementary Fig. 8). Next, we incorporated furan into peptides by coupling 2-furylalanine (A$_{fur}$) through solid-phase synthesis and carried out the selective modification of 2-furylalanine on unprotected peptides, Ac-(A$_{fur}$)NF **1p** and Ac-FN(A$_{fur}$)AFK **1q** under optimized FuTine MCR conditions to generate *N*-pyrrole-modified peptides **2p** and **2q** in good yields (87% and 53%, respectively, Fig. 4b and Supplementary Fig. 9). These examples showed the high precision of this reaction in selectively modifying Lys, Cys, or furan irrespective of the presence of other amino acids.

## Application of FuTine MCR in peptide macrocyclization and stapling

Since the reaction between cysteine and lysine in the presence of oxidized furan is orthogonal, we explored this chemistry for the synthesis of macrocyclic peptides (**2r-2t**) of varying amino acid composition and ring sizes (15-32) by carrying out FuTine reaction on unprotected linear peptides Ac-CFK **1r**, Ac-CNAFK **1s**, and Ac-CADYSRFK **1t** containing cysteine and lysine at the two termini (Fig. 4c and Supplementary Fig. 10). The incubation of linear peptides **1r-1t** with 1.2 equiv. of oxidized furan in ACN:H$_2$O (5:1) afforded macrocycles **2r-2t** with an *N*-pyrrole moiety at the site of cyclization in high conversions (78->99%), without the formation of dimers under the reaction conditions (Fig. 4c and Supplementary Fig. 10). Similar attempts to synthesize macrocycles by reaction between His and Lys and between Arg and Lys did not work under the reaction conditions (Supplementary Fig. 10). These experiments further confirm the orthogonal nature of FuTine chemistry between Cys and Lys. To determine the preference for the formation of particular ring size during macrocyclization, we carried out FuTine reaction with linear peptides FMKNYC **1u** and Ac-KFMKNYC **1v**, containing either one N-terminus and one lysine or two reactive lysines, respectively. The results of cyclization on these peptides did not show any preference in regards to the ring sizes and both the macrocycles (**2u** and **2u'**) and (**2v** and **2v'**) are formed with almost similar conversions (Fig. 4d and Supplementary Fig. 10). We do not expect to see similar observation with proteins, where different lysines have different microenvironments that influence their pKa. Thus, there would be a preference for a

particular lysine residue over the others in proteins. The site of the macrocyclization was determined by cleaving the cyclic peptides (**2u** and **2u'**) and (**2v** and **2v'**) at Met using CNBr, and the resulting fragments were analyzed by HRMS (Supplementary Fig. 10). This study demonstrated the robust nature of FuTine method for the synthesis of peptide macrocycles.

Encouraged by the modification of both the amines of the diamine 1,3-bis(aminomethyl)benzene to dual *N*-pyrrole moieties in one pot (**2m**, Fig. 3a), we extended its application for stapling of peptides between two cysteine or two lysine residues. To achieve this goal, we subjected a peptide Ac-HCSNCF **1w** containing two cysteine residues at *i* and *i + 3* positions to 1,3-bis(aminomethyl)benzene in the presence of oxidized furan to generate stapled peptide **2w** with two *N*-pyrrole groups in high conversion (>99%, Fig. 4e and Supplementary Fig. 11). Next, we carried out stapling of a peptide Ac-GKVAKF **1x** containing two lysine residues at *i* and *i + 3* positions with 1,4-butanedithiol and observed full conversion to a stapled peptide **2x** with two *N*-pyrrole rings (>99%, Fig. 4e, Supplementary Fig. 12).

## Chemoselective protein modification by FuTine MCR

We next evaluated FuTine MCR for the chemoselective modification of lysine residues on proteins. We initiated our experiments using a model protein myoglobin containing 16 lysines by varying the equivalents of oxidized furan (1–15 equiv.) and thioglycolic acid (1–15 equiv.). Notably, we were able to efficiently modify 63% of myoglobin even with 1 equiv. of the oxidized furan and thioglycolic acid at very low concentrations (40 μM) (Fig. 5a and Supplementary Fig. 13). High equivalents (15 equiv.) of furan and thiol quantitatively modified >99% myoglobin with the maximum conversion of six-modified lysines along with the modification of up to 10 lysines on myoglobin.

The optimized conditions for the modification of a protein were determined to be 5 equiv. of oxidized furan and 5 equiv. of thioglycolic acid in 10 mM phosphate buffer (pH 7.5) at room temperature for 16 h. Under optimized conditions, we were able to efficiently modify >99% of myoglobin at very low concentrations (40 μM) with a maximum yield of three-modified lysines (46%) along with the modification of up to 5 lysines. Although there are 16 lysines on myoglobin, only a few underwent modifications at low equivalents of reagents. This result points to the potential application of this approach for activity-based protein profiling (ABBP) by identifying reactive lysines in the active site and distinguishing them from less reactive lysines. Next, we carried out MS/MS analysis on a myoglobin sample that was treated with 1 equiv. of oxidized furan and thioglycolic acid and determined the sites of modification on myoglobin to be K79 and K63 in the ratio of 3:1 (Fig. 5b and Supplementary Fig. 13). No modification of myoglobin was observed with oxidized furan in the absence of thiol, as was observed with the small molecule experiments, further confirming that all three components (furan, thiol and amine) are required for the efficient modification of proteins. After purification, incubation of modified myoglobin over a pH range of 3 to 11 at 37 °C showed full product stability of *N*-pyrrole over a 24 h period (Supplementary Fig. 14).

## Protein function is retained after modification

To explore the ability of our method to modify a protein without altering its tertiary structure, we examined the activity of pyrrole-modified myoglobin (Pyr-Myo) with three modifications by its ability to carry out the oxidation of *o*-phenylenediamine with hydrogen peroxide to 2,3-diaminophenazine as monitored at 426 nm by UV spectroscopy[30]. Negligible change in the UV signal was observed as compared to unmodified myoglobin (Fig. 5c and Supplementary Fig. 15). We also carried out the circular dichroism (CD) studies with both modified and unmodified myoglobin and observed no change in the secondary structure after the modification (Fig. 5c and

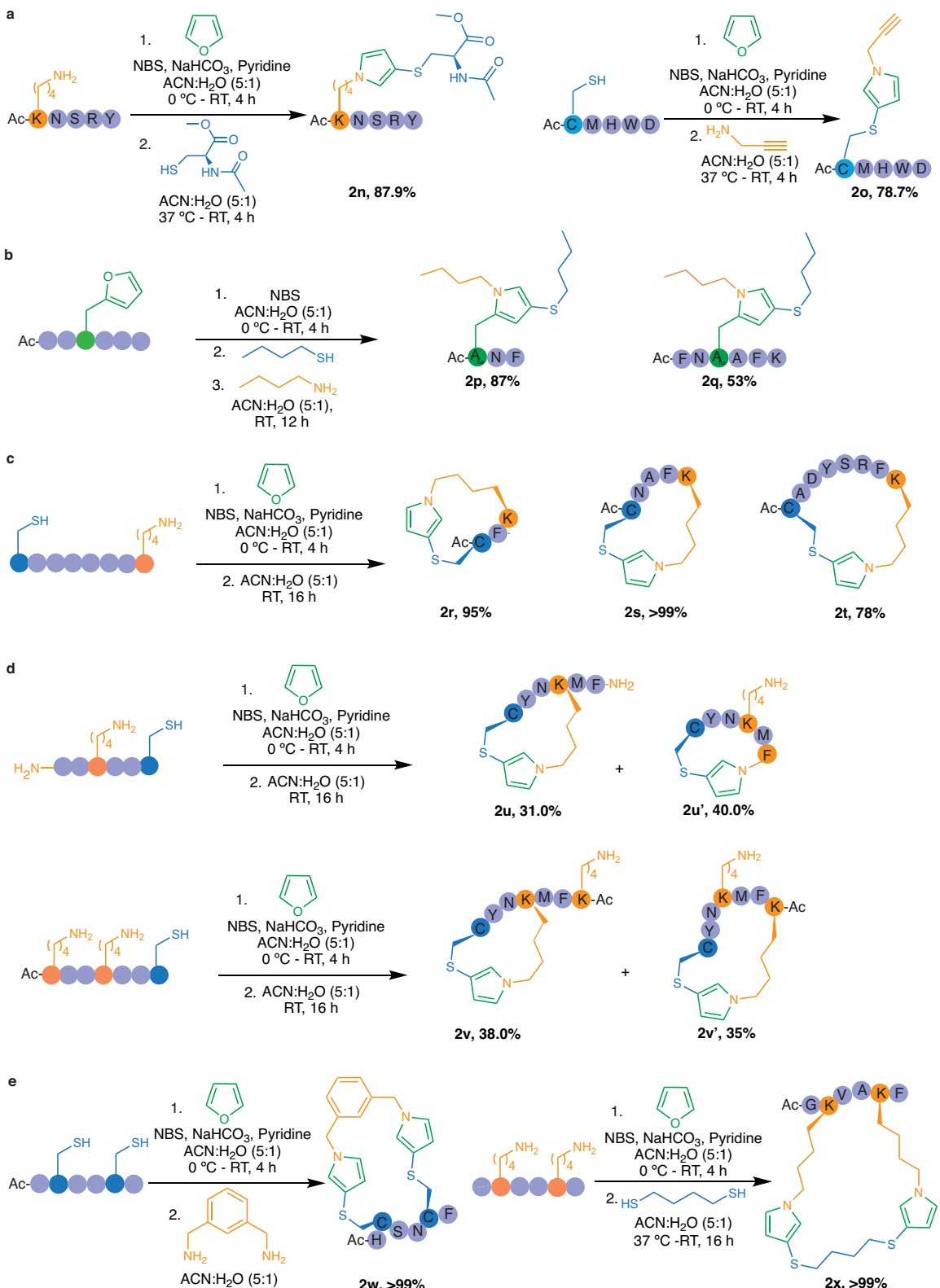

**Fig. 4 | Furan-thiol-amine (FuTine) multicomponent reaction (MCR) for chemoselective modification and cyclization of peptides. a** Chemoselective modification of lysine or cysteine to form *N*-pyrrole on peptides without modification of any reactive amino acids. **b** Chemoselective modification of 2-furylalanine to form *N*-pyrrole on peptides. **c** Synthesis of peptide macrocycles of varying ring sizes

between Cys and Lys side chains by the addition of oxidized furan.
**d** Macrocyclization showed no preference for a ring size, N-terminus, and reactive lysine residue. **e** Stapling of peptides between two cysteine or two lysine residues on peptides by the addition of diamine or dithiol and oxidized furan.

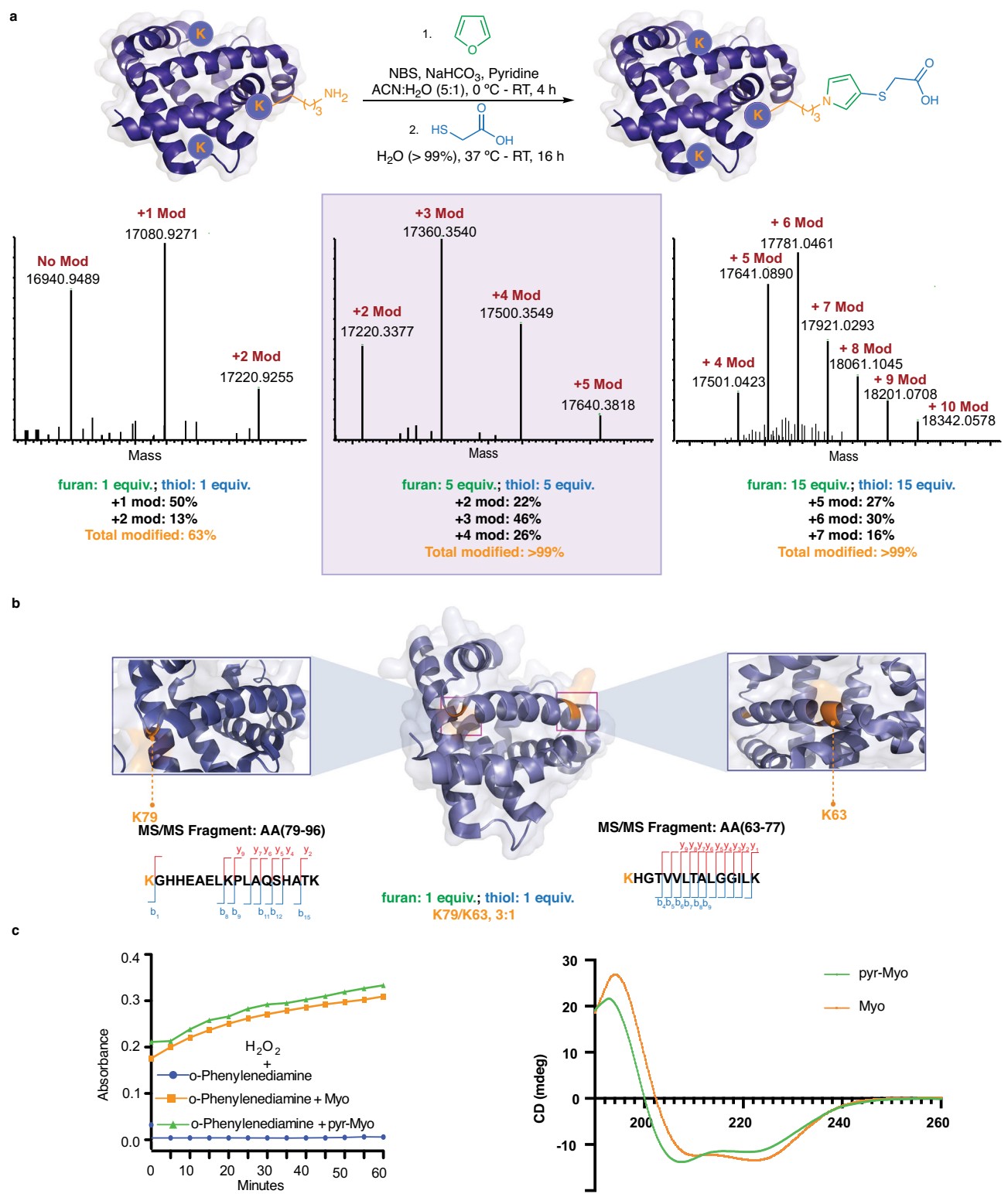

**Fig. 5 | One-pot furan-thiol-amine (FuTine) multicomponent reaction (MCR) for chemoselective modification of proteins. a** Optimization of the selective modification of lysine on protein myoglobin by varying equivalents of both oxidized furan (1–15 equiv.) and thiol (1–15 equiv.) reagents. The full conversion was observed by using 5 equiv. of thiol and furan indicating high robustness and efficiency of the thiol-amine reaction. **b** MS/MS analysis of modified myoglobin (treated with 1 equiv. each of thiol and furan) showed the modification of K79 and K63 in a 3:1 ratio. **c** pyrrole-modified myoglobin (pyr-Myo) under the optimized conditions (treatment with 5 equiv. of thiol and furan) demonstrates a similar ability to oxidize o-phenylenediamine as compared to unmodified myoglobin. This data supports the hypothesis that the 3D structure of the myoglobin remained intact after the modification, which was further verified by circular dichroism analysis.

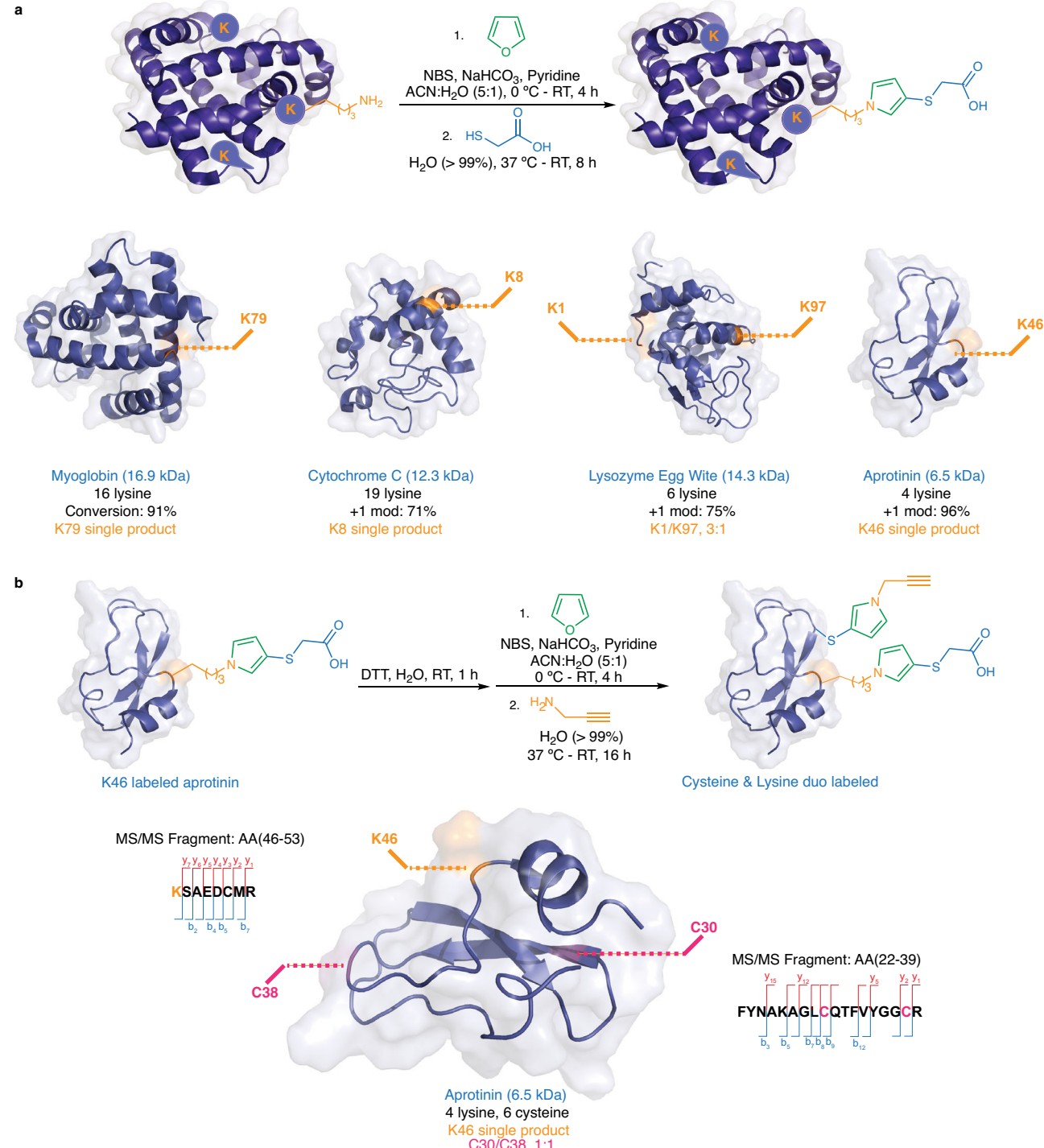

**Fig. 6 | Homogeneous labeling of proteins using furan-thiol-amine (FuTine) chemistry. a** Optimized homogeneous labeling conditions (1.2 equiv. of *cis*−2-butene-1,4-dial (BDA) and 1.2 equiv. of thioglycolic acid, 8 h, RT) of lysine residues on four different proteins. **b** Dual labeling of cysteine and lysine residues on aprotinin with MS/MS analysis showed modification of K46, C30 and C38. Created with BioRender.com.

Supplementary Fig. 15). These results highlight the ability of FunTine MCR to enable efficient and selective modification of proteins without denaturation, conserving their structure and bioactivity.

## Application of FuTine MCR in generating homogeneous proteins

In proteomics, homogeneous labeling of lysine residues is a significant challenge. Inspired by our initial screening on myoglobin (Fig. 5a), we attempted to use 1.2 equiv. of thiol and BDA with a combination of

shorter reaction time (8 h) and higher protein concentration in reaction to achieve homogeneous labeling of lysine (Fig. 6a and Supplementary Fig. 16). We carried out this optimized protocol on four different commercially available protein substrates (myoglobin, cytochrome C, lysozyme egg white and aprotinin) and observed homogeneous labeling in all cases. We determined the sites of modification on proteins using MS/MS sequencing (Supplementary Fig. 16). In contrast, literature reports of lysine modification with well-known NHS-ester led to the non-specific labeling of other amino acids such as

Ser and Arg[31]. In our hand, we also observed the formation of a heterogeneous mixture of products on myoglobin with NHS-ester under the reported reaction conditions (Supplementary Fig. 16). These results indicated that our method is specific for labeling lysine and is amenable to generate homogeneous products in high conversions. Previous methods for labeling lysine either generated a heterogeneous mixture of products modifying multiple lysine residues[32,33] or form homogeneous products mediated by heterobifunctional molecules[34–36]. In the later cases, the length of linkers between two different functional groups dictated the regioselectivity, rather than the inherent reactivity of the lysine residues on a protein[35,36]. Such methods have not been reported for chemoproteomic profiling of lysine.

Next, we carried out dual labeling of cysteine and lysine separately using FuTine chemistry on aprotinin (Fig. 6b and Supplementary Fig. 17). We first homogeneously labeled the K46 residue on aprotinin followed by the disulfide bond cleavage using dithiothreitol. We then carried out FuTine labeling of cysteine residues using 5 equiv. of BDA and amine, giving dual labeled aprotinin on cysteine (C30 and C38) and lysine (K46) as confirmed by MS/MS sequencing. There is no current methodology in the literature that can target both cysteine and lysine residue together on proteins using the same chemistry.

## Application of FuTine MCR in protein modifications

Next, we applied FuTine MCR for modification of a broad range of commercially available protein substrates (insulin, ubiquitin, cytochrome C, carbonic anhydrase, alpha-lactalbumin, lysozyme and others) with a wide range of molecular weights (5,000 Da to > 79,000 Da) and varying three-dimensional structures affording >99% modification with excellent selectivity for lysine at low concentrations (8 to 114 μM) (12 examples, Fig. 7 and Supplementary Fig. 18). The mild conditions of this reaction maintained the structurally critical disulfide bonds in insulin (Fig. 7). Encouraged by the high labeling efficiency of FuTine MCR, we showcased its application in the selective labeling of native proteins with a fluorophore. We incubated native proteins BSA, creatine kinase and transferrin with oxidized furan and thiol-FITC for 16 h at room temperature followed by protein precipitation and analysis by sodium dodecyl-sulfate polyacrylamide gel electrophoresis (SDS-PAGE) using in-gel fluorescence. The results clearly showed the labeling of BSA, creatine kinase and transferrin with fluorophores in the presence of all three reaction components (lanes 4, Fig. 8a and Supplementary Fig. 19). No fluorophore labeling was observed in control experiments in the absence of any one component (lanes 1–3, Fig. 8a). We next utilized FuTine chemistry for the selective fluorescent labeling of cysteine on the same three native proteins, BSA, creatine kinase and transferrin. We incubated the proteins with dithiothreitol (DTT) for 1 h to cleave the disulfide bonds and incubated the sample with oxidized furan and AZ680-amine dye (NH$_2$-AZ680) for 16 h at room temperature followed by protein precipitation and analysis by SDS-PAGE using in-gel fluorescence. The results clearly showed the labeling of BSA, creatine kinase and transferrin with fluorophores in the presence of all the three reaction components (lanes 4, Fig. 8a and Supplementary Fig. 19). No fluorophore labeling was observed in control experiments in the absence of any one component (lanes 1–3, Fig. 8a).

Due to the high efficiency and robustness of FuTine chemistry to form a staple between two lysine residues on peptides, we explored the applicability of this reaction for the selective in situ cyclization of proteins between two lysine residues to increase the stability of protein tertiary structures[37]. The current protein cyclization methods mainly rely on the addition of unnatural amino acids, cysteine residues and enzyme-recognized peptide tags at particular positions in proteins[38]. To achieve protein cyclization, we incubated native myoglobin with oxidized furan generated in situ and 1,4-butanedithiol and observed the crosslinking between two Lys via dual N-pyrrole moieties

in 38% conversion (Fig. 8b and Supplementary Fig. 20). These results showed the potential application of this chemistry for the selective modification of native proteins and protein cyclization with enhanced stability towards chemical denaturation while retaining catalytic activity. Although there are several electrophiles available for labeling nucleophilic lysine such as activated esters[39], sulfonyl chlorides[40] or isothiocyanates[41], these are limited due to cross-reactivity with other reactive amino acids on proteins. Modification of lysine by reductive alkylation does not stop after single addition and leads to the formation of varying products due to multiple alkylations on a single lysine[42]. There are no other bioconjugation reactions known in the literature for achieving such high efficiency (>99%) and selective modification of lysine and protein cyclization, independent of the size and 3D structure of proteins under such mild and dilute reaction conditions (~8 μM). The high precision and efficiency of FuTine MCR in selectively modifying Cys or Lys is unique and expands its scope for multiple applications including the dual modification of proteins with varying functional tags.

The incorporation of two different types of functionalities at distinct locations on a protein greatly expands the features of native proteins. We first selectively modified transferrin by the selective labeling of tyrosine with an alkyne-derived phenyl diazonium salt to generate a diazo-complex with tyrosine. The alkyne group was further labeled with azide-Cy5 dye using click chemistry resulting in the labeling of transferrin with Cy5 fluorophore. Next, lysine was selectively labeled with oxidized furan and thiol-FITC to generate lysine-substituted N-pyrrole-FITC resulting in the dual labeling of transferrin with two different fluorophores. We analyzed the dual labeling of transferrin by in-gel fluorescence analysis showing the modification of a protein with two different dyes, FITC and Cy5 (lane 4, Fig. 9a and Supplementary Fig. 21). This dual labeling of proteins showed that our reaction is orthogonal, highly chemoselective and compatible with other labeling techniques. This result highlighted the importance of the synergistic/additive properties of the two synthetic moieties on dual-modified protein bioconjugates and provides avenues to expand its applications in introducing FRET donor-acceptor dyes for analysis of protein structures, protein-protein interactions and combining drug molecules with imaging agents to design precision protein therapeutics[43–45].

## Labeling of proteins in complex cell lysate mixture

Due to the high efficiency of this reaction in labeling proteins with full conversion independent of protein size and 3D structure, we further explored its ability to label proteins in a complex cell lysate mixture. Since FuTine MCR is highly chemoselective for labeling either lysine or cysteine on proteins, we showcased the application of this chemistry for the modification of both lysine and cysteine on proteins in a complex cell lysate mixture by changing the identity of the exogenous nucleophile. For labeling lysine, cell lysate was incubated with oxidized furan and thiol-FITC, and for labeling cysteine, cell lysate was incubated with oxidized furan and AZ680-amine dye (NH$_2$-AZ680). Both reactions were left for 4 h at room temperature and the proteins were precipitated for analysis by in-gel fluorescence. The results clearly showed extensive labeling of lysine or cysteine on proteins in a complex cell lysate mixture in the presence of all three components (lane 4, Fig. 9b and Supplementary Fig. 22). No labeling was observed in the control experiments in the absence of any one of the components (lanes 1–3, Fig. 9b). To further confirm our claims on targeting either lysine or cysteine in the proteome by using FuTine chemistry. We have carried out a competition inhibition assay in both cases. We used iodoacetamide, a well-known cysteine-selective probe, in various concentrations to block free cysteine residues on cell lysate. We then incubated the treated cell lysate sample with oxidized furan and AZ680-amine dye (NH$_2$-AZ680). The proteins were precipitated for analysis by in-gel fluorescence, which clearly showed differences in

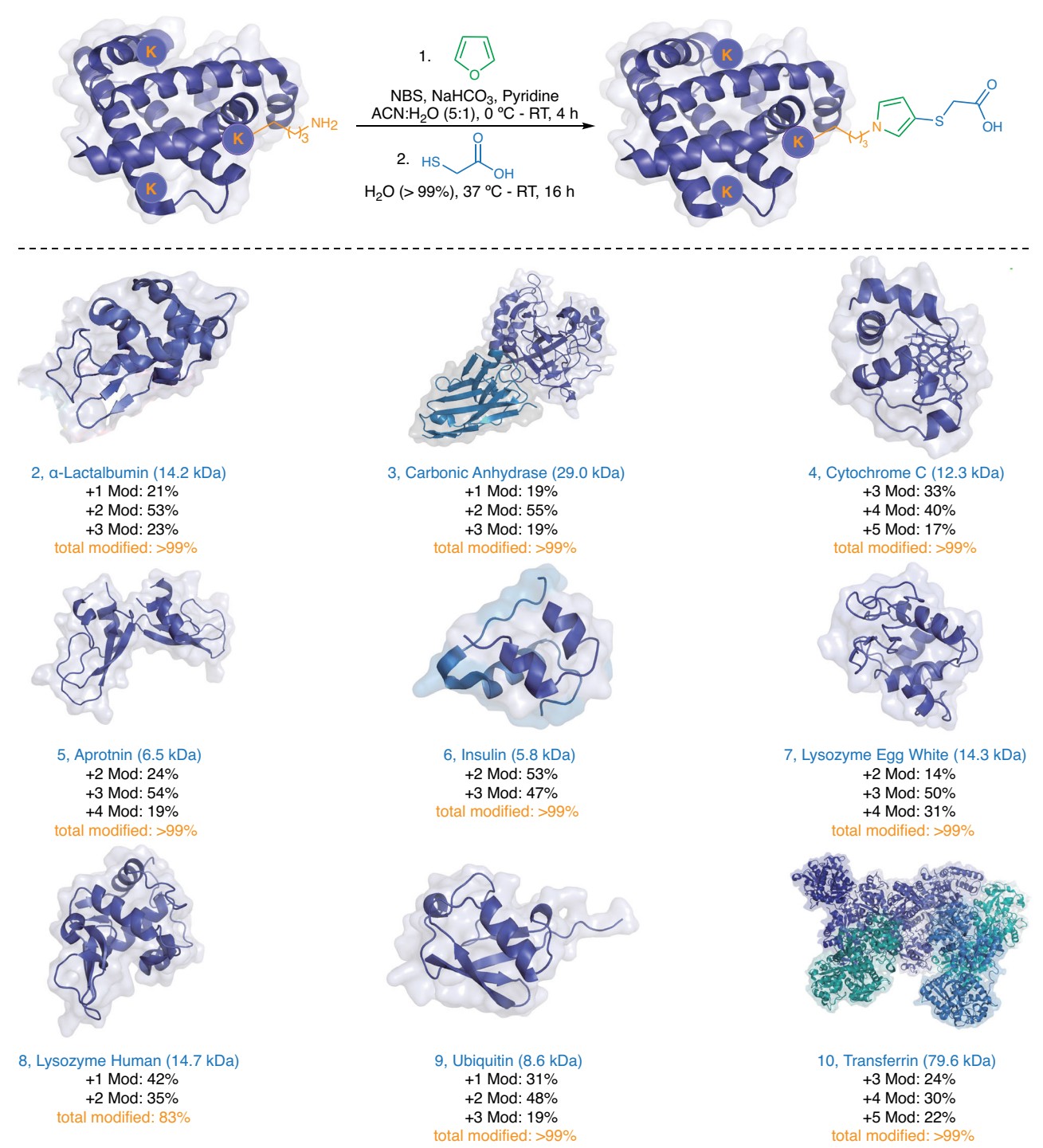

**Fig. 7 | Application of furan-thiol-amine (FuTine) multicomponent reaction (MCR) for chemoselective modification of diverse proteins.** Chemoselective modification of lysine on proteins of varying sizes (5–79 KDa) and 3D-structures with >99% conversion in most of cases under optimized conditions analyzed by MS. Created with BioRender.com.

florescence intensity between samples that were treated with iodoacetamide and samples that weren't in a dose-dependent manner (Supplementary Fig. 22). For targeting lysine, we incubated cell lysate with different concentrations of NHS-ester followed by treatment with oxidized furan and thiol-FITC. As expected, we observed differences in fluorescence intensity between samples that were treated with NHS-ester and samples that weren't in a dose-dependent manner (Supplementary Fig. 22). This assay confirmed the selectivity of FuTine chemistry for lysine and cysteine and the ability of FuTine chemistry for carrying out chemoproteomic profiling of cysteine and lysine residues in a complex cell lysate.

## Discussion

Inspired by the enzyme catalyzed toxicity of furan generating cross-linked proteins inside cells, we developed a robust and highly efficient one-pot multicomponent chemoselective reaction for coupling thiols and amines with furans to generate highly stable *N*-pyrrole products. The reactivity of this protocol is

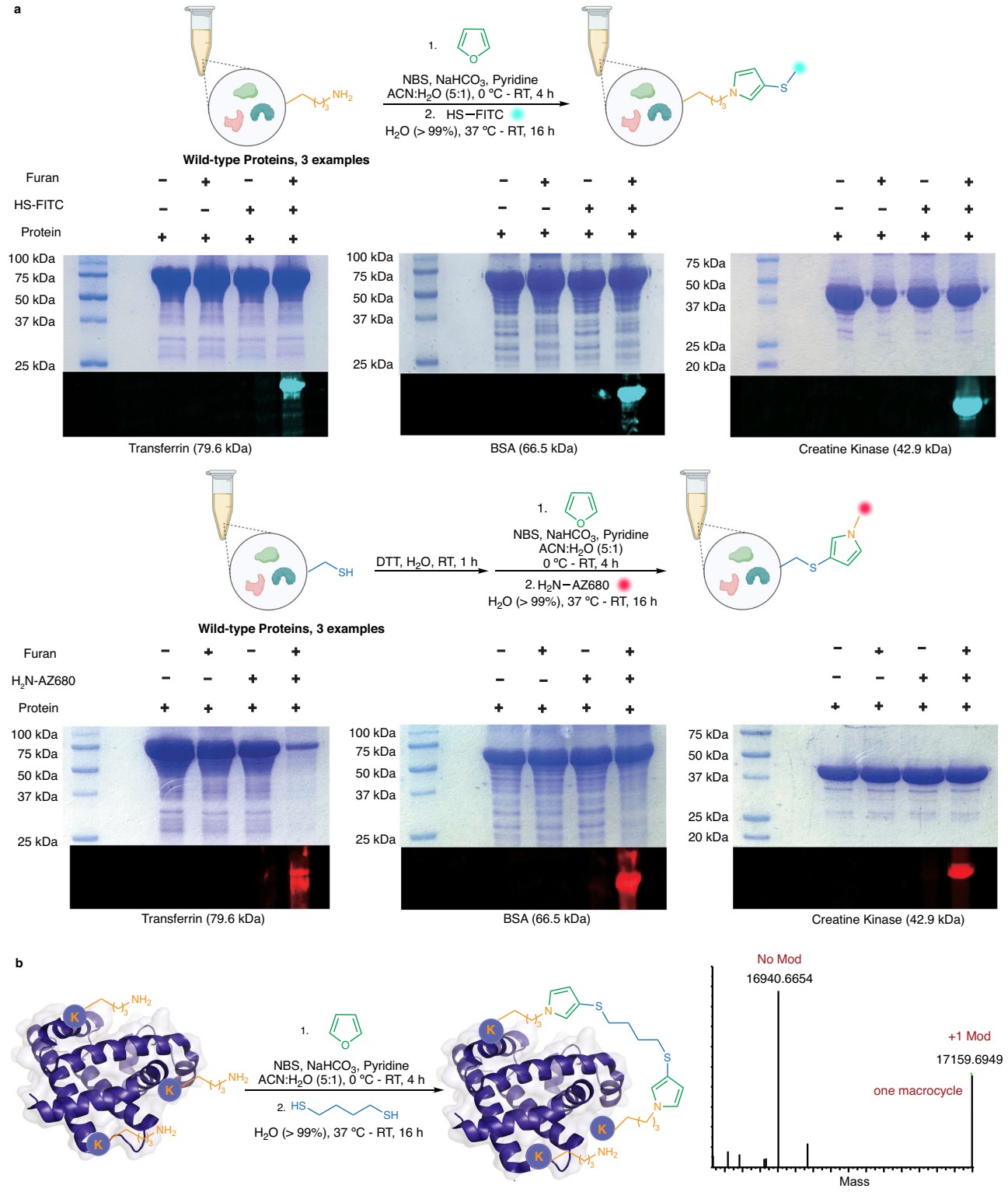

**Fig. 8 | Application of furan-thiol-amine (FuTine) reaction for fluorophore labeling of proteins and cyclization of proteins. a** Fluorophore labeling of lysine and cysteine in native proteins and their analysis by in-gel fluorescence. Independent gels were run twice for each experiment of labeling lysine and cysteine.

**b** Macrocyclization of myoglobin between two lysine residues by oxidized furan and 1,4-butanedithiol leading to a single monocyclic product. Created with BioRender.com.

unique because of the selectivity of the oxidized furan to first add a single thiol followed by the addition of a single amine to generate a chromophoric thio-*N*-pyrrole heterocycle as the only product under mild conditions and without any catalysts. The reaction exhibits high substrate scope that provides clean conversations to stable *N*-pyrrole products independent of the functional groups present on the coupling units. We further derivatized the *N*-pyrrole product via nucleophilic addition on alkynedicarboxylates to incorporate diverse functional tags. We demonstrate the application of the FuTine MCR for the selective

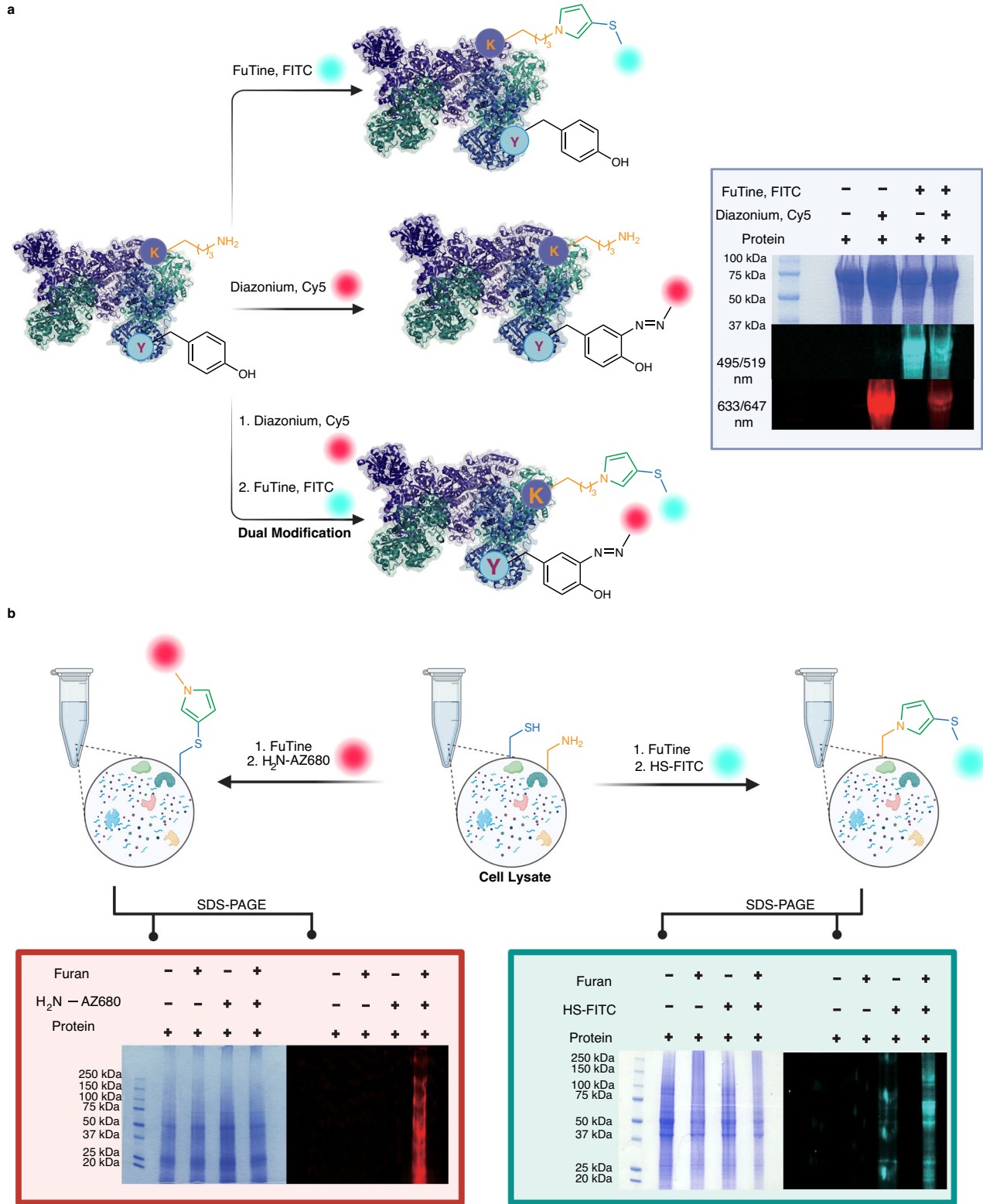

**Fig. 9 | Application of furan-thiol-amine (FuTine) multicomponent reaction (MCR) in dual labeling of proteins and modification of proteins in cell lysate.** **a** Dual modification of a protein with two different fluorophores by two different strategies. First, selective labeling of tyrosine via phenyl diazonium salts generating diazo complexes followed by selective labeling of lysine via FuTine MCR. Independent gels were run twice for dual labeling of a protein. **b** Selective labeling of lysine or cysteine on proteins in a cell lysate using thiol-FITC or AZ680-amine dye (NH₂-AZ680) and oxidized furan as analyzed by in-gel fluorescence analysis (for lysine labeling-right side gel; for cysteine labeling-left side gel, lane 4). The reaction works only in the presence of all three components as no fluorophore labeling was observed with thiol or amine in the absence of oxidized furan (lanes 1–3). Independent gels were run twice for labeling cell lysate. Created with BioRender.com.

modification of peptides, and synthesis of macrocyclic and stapled peptides. This reaction was successfully applied for the modification of twelve different proteins of varying sizes and 3D structures with high efficiency. The resulting pyrrole-protein conjugates showed high stability in varying acidic and basic pH conditions. In addition, FuTine chemistry is compatible with other conjugation chemistry as shown by the post-modification of a tyrosine residue by diazo coupling. We demonstrate the application of FuTine chemistry to generate homogeneous protein conjugates, and its ability to distinguish subtle reactivity differences among lysines on native proteins is remarkable for protein engineering. We utilized FuTine chemistry for the selective labeling of lysine and cysteine on proteins with fluorophores in a complex cell lysate mixture. The reaction was utilized for the dual labeling of both lysine and cysteine on a protein in a selective manner by controlling the amounts of amine and thiol reagents. The reaction was deployed successfully for the homogeneous stapling of a protein between two lysine residues. Our discovery of harnessing FuTine chemistry for the precise homogeneous protein engineering, homogeneous stapling of proteins and chemoproteomic profiling in a complex cell lysate is a significant advance in protein modification. The majority of lysine modification approaches either typically generate a heterogeneous mixture of products modifying multiple lysine residues or a few generate homogeneous products typically requiring the heterobifunctional molecules joined by linkers. One of the drawbacks of the FuTine method for the modification of peptides and proteins is the low stability of BDA, so it needs to be prepared freshly for each reaction. Secondly, both furan and thiol are sensitive to oxidizing conditions therefore BDA needs to be generated separately followed by the addition of thiol and amine for obtaining a high yield of the conjugate. Overall, this approach provides a multicomponent toolbox that will be applicable to many fields of chemistry.

## Methods

### General procedure for furan-thiol-amine multicomponent reaction

Furan **1** (500 µL, 6.88 mmol, 1 equiv.) and sodium bicarbonate (575 mg, 6.88 mmol, 1 equiv.) were added to a solution of 60 mL $CH_3COCH_3$:$H_2O$ (5:1). The reaction mixture was cooled to 0 °C and left to stir for 15 min. N-Bromosuccinimide NBS (1224 mg, 6.88 mmol, 1 equiv.) was dissolved in a solution of 60 mL $CH_3COCH_3$:$H_2O$ (5:1) and added to the reaction mixture dropwise. After the addition of NBS, the reaction mixture was left to stir for 10 min. Pyridine (1110 µL, 13.76 mmol, 2 equiv.) was then added to the reaction mixture, which was allowed to stir for 4 h at RT and then used without further purification. The thiol derivative (4 equiv.) was then added, and the reaction mixture was incubated at 37 °C for 30 min before the addition of the amine derivative (2 equiv.). The reaction mixture was then left to stir for 16 h at 60 °C. The volatiles were removed under reduced pressure and separated by work-up using ethyl acetate and 1 M aqueous HCl (20:1). The organic layer was extracted, dried with $Na_2SO_4$, and the crude residue was further purified by column chromatography.

### General procedure for modification of 3-thio-N-pyrrole

**5a**. 1-butyl-3-(butylthio)-1*H*-pyrrole (0.47 mmol, 1 equiv.) was dissolved in 10 mL of acetonitrile and added to a reaction mixture of dimethyl acetylenedicarboxylate (1.5 equiv.) in acetonitrile (10 mL) and the reaction mixture was stirred for 16 h at RT. TLC was used to monitor the reaction progression. The volatiles were removed under reduced pressure and the crude product was purified by column chromatography using a mixture of hexanes and ethyl acetate (7:3) as the eluent

to get product **5a**. **5a** was obtained in 49.7% yield as a bright yellow oil. **5a** was characterized by $^1H$ and $^{13}C$ NMR.

### Procedure for modification of linear peptides containing lysine

Furan **1** (100 µL, 1.38 mmol, 1 equiv.) and sodium bicarbonate (115 mg, 1.38 mmol, 1 equiv.) were added to a solution of 12 mL ACN:$H_2O$ (5:1). The reaction mixture was cooled to 0 °C and left to stir for 15 min. N-Bromosuccinimide NBS (244 mg, 1.38 mmol, 1 equiv.) was dissolved in a solution of 12 mL ACN:$H_2O$ (5:1) and added to the reaction mixture dropwise. Afterward, the reaction mixture was left to stir for 10 min, and pyridine (222 µL, 2.76 mmol, 2 equiv.) was added to the reaction mixture. The reaction mixture was stirred for 4 h and used without further purification. From the pot, 1.5 equiv. (184 µL) of the reaction solution was taken and incubated with N-acetyl-cysteine methyl ester (7.4 mg, 6 equiv.) at 37 °C for 30 min in 5 mL of acetonitrile and 1 mL of water. 5 mg of Ac-KNSRY (**1n**, 1 equiv.) was dissolved in 6 mL of ACN:$H_2O$ (5:1) and was then added to the reaction mixture. The concentration of peptide in the reaction mixture was 589 nM. The reaction mixture was left to stir for 4 h at room temperature. Samples were injected directly into liquid chromatography-mass spectrometry (LC-MS) to analyze the peptide modification. The reaction mixture was analyzed by HPLC using method A to determine the percent conversion to modified product **2n** (87.9%).

### Procedure for modification of linear peptide containing cysteine

Furan **1** (100 µL, 1.38 mmol, 1 equiv.) and sodium bicarbonate (115 mg, 1.38 mmol, 1 equiv.) were added to a solution of 12 mL ACN:$H_2O$ (5:1). The reaction mixture was cooled to 0 °C and left to stir for 15 min. N-Bromosuccinimide (244 mg, 1.38 mmol, 1 equiv.) was dissolved in a solution of 12 mL ACN:$H_2O$ (5:1) and added to the reaction mixture dropwise. Afterward, the reaction mixture was left to stir for 10 min, and pyridine (222 µL, 2.76 mmol, 2 equiv.) was added to the reaction mixture. The reaction mixture was stirred for 4 h and used without further purification. 1.5 equiv. (68 µL) of the reaction mixture was taken from a pot and incubated with 1.9 mg of Ac-CMHWD (**1o**, 1 equiv.) in 6 mL of ACN:$H_2O$ (5:1) at 37 °C for 30 min. The concentration of peptide in the reaction mixture was 1.13 µM. Propargylamine (1.2 equiv.) was then added to the reaction mixture and left to stir for 4 h at room temperature. The reaction mixture was analyzed by HPLC using method A to determine the percent conversion to modified product **2o** (78.7%).

### Procedure for the modification of linear peptide containing furan

1.5 mg of peptide **1p** (3.3 µmol, 1 equiv.) was dissolved in 2 mL of ACN:$H_2O$ (5:1). NBS (1.8 mg, 9.3 µmol, 3 equiv.) was added in one portion at 0 °C and left to stir at RT for 4 h. 1-butanethiol (2.1 µL, 19.7 µmol, 6 equiv.) and 1-butylamine (1.6 µL, 16.4 mmol, 5 equiv.) were then added from prepared stock solutions in ACN to the reaction vial. The solution was stirred at RT for 12 h. Following the completion of the reaction, the solvent was removed using a centrifugal vacuum concentrator system. The reaction mixture was re-dissolved in 350 µL of ACN:$H_2O$ (5:1) and was analyzed by HPLC using method A to determine the percent conversion to modified product **2p** (87%).

### General procedure for the macrocyclization of linear peptides

A solution of furan **1** (100 µL, 1.4 mmol, 1 equiv.) and sodium bicarbonate (118 mg, 1.4 mmol, 1 equiv.) in 12 mL of ACN:$H_2O$ (5:1) was incubated at 0 °C for 15 min. A 12 mL solution of NBS (244 mg, 1.4 mmol, 1 equiv.) in ACN:$H_2O$ (5:1) was added to the mixture dropwise over 5 min at 0 °C and the reaction was stirred for 10 min. Pyridine (222 µL, 2.8 mmol, 2 equiv.) was then added directly and the reaction mixture was allowed to stir at RT for 4 h and used without further purification. 1.2 equiv. (47 µL) of the reaction mixture was taken from a

pot and incubated with 1 mg of peptide 1r (2.3 μmol, 1 equiv.) in a 6 mL solution of ACN:H$_2$O (5:1) and the reaction was allowed to stir at RT for 16 h. Following the completion of the reaction, the solvent was removed using a centrifugal vacuum concentrator system. The reaction mixture was re-dissolved in 350 μL of ACN:H$_2$O (5:1) and was analyzed by HPLC using method A to determine the percent conversion to product 2r (95%).

## Procedure for stapling of peptides between two cysteine residues

Furan 1 (100 μL, 1.38 mmol, 1 equiv.) and sodium bicarbonate (115 mg, 1.38 mmol, 1 equiv.) were added to a solution of 12 mL ACN:H$_2$O (5:1). The reaction mixture was cooled to 0 °C and left to stir for 15 min. N-Bromosuccinimide (244 mg, 1.38 mmol, 1 equiv.) was dissolved in a solution of 12 mL ACN:H$_2$O (5:1) and added to the reaction mixture dropwise. Afterward, the reaction mixture was left to stir for 10 min, and pyridine (222 μL, 2.76 mmol, 2 equiv.) was added to the reaction mixture. The reaction mixture was stirred for 4 h and used without further purification. .2.5 equiv. (15 μL) of this reaction mixture was taken directly from the pot and incubated with 0.25 mg of Ac-HCSNCF (1u, 1 equiv.) in 600 μL of ACN:H$_2$O (5:1) at 37 °C for 30 min. m-Xylenediamine (1 equiv.) was dissolved in 600 μL of ACN:H$_2$O (5:1) and added to the reaction mixture dropwise for a period of 30 min. The concentration of peptide in the reaction mixture was 556 μM. The reaction mixture was left to stir for 16 h at RT. The reaction mixture was analyzed by HPLC using method A to determine the percent conversion to modified product 2w (>99%).

## Procedure for stapling of peptides between two lysine residues

Furan 1 (100 μL, 1.38 mmol, 1 equiv.) and sodium bicarbonate (115 mg, 1.38 mmol, 1 equiv.) were added to a solution of 12 mL ACN:H$_2$O (5:1). The reaction mixture was cooled to 0 °C and left to stir for 15 min. N-Bromosuccinimide (244 mg, 1.38 mmol, 1 equiv.) was dissolved in a solution of 12 mL ACN:H$_2$O (5:1) and added to the reaction mixture dropwise. Afterward, the reaction mixture was left to stir for 10 min, and pyridine (222 μL, 2.76 mmol, 2 equiv.) was added to the reaction mixture. The reaction mixture was stirred for 4 h and used without further purification. 2.5 equiv. (63 μL) of the reaction mixture was taken from the pot and incubated with 1,4-butanedithiol (1 equiv.) at 37 °C for 30 min. 1 mg of GKVAKF (1v, 1 equiv.) was dissolved in a 2 mL solution of ACN:H$_2$O (5:1) and added to the reaction mixture. The concentration of peptide in the reaction mixture was 726 μM. The reaction mixture was left to stir for 16 h at RT. The reaction mixture was analyzed by HPLC using method A to determine the percent conversion to modified product 2x (>99%).

## Procedure for the modification of proteins

Furan 1 (100 μL, 1.38 mmol, 1 equiv.) and sodium bicarbonate (115 mg, 1.38 mmol, 1 equiv.) were added to a solution of 12 mL ACN:H$_2$O (5:1). The reaction mixture was cooled to 0 °C and left to stir for 15 min. N-Bromosuccinimide (244 mg, 1.38 mmol, 1 equiv.) was dissolved in a solution of 12 mL ACN:H$_2$O (5:1) and added to the reaction mixture dropwise. Afterward, the reaction mixture was left to stir for 10 min, and pyridine (222 μL, 2.76 mmol, 2 equiv.) was added to the reaction mixture. The reaction mixture was stirred for 4 h and used without further purification. From the pot, 1.2 equiv. (2.5 μL) of the mixture was taken and incubated with thioglycolic acid (1.2 equiv.) at 37 °C for 30 min in 1 mL of water. 2 mg of myoglobin (1 equiv.) was dissolved in 2 mL of water and added to the reaction mixture. The reaction mixture was left to stir for 16 h at RT. The reaction mixture was purified by molecular weight cut-off and characterized by LC-MS to analyze the protein modification. The modification of myoglobin was repeated with varying amounts of furan and thiol reagents to identify optimized conditions for selective protein labeling. Percent conversions were calculated based on the deconvolution spectra.

## Procedure for cyclization of proteins

Furan 1 (100 μL, 1.38 mmol, 1 equiv.) and sodium bicarbonate (115 mg, 1.38 mmol, 1 equiv.) were added to a solution of 12 mL ACN:H$_2$O (5:1). The reaction mixture was cooled to 0 °C and left to stir for 15 min. N-Bromosuccinimide (244 mg, 1.38 mmol, 1 equiv.) was dissolved in a solution of 12 mL ACN:H$_2$O (5:1) and added to the reaction mixture dropwise. Afterward, the reaction mixture was left to stir for 10 min, and pyridine (222 μL, 2.76 mmol, 2 equiv.) was added to the reaction mixture. The reaction mixture was stirred for 4 h and used without further purification. From the furan pot, 2 equiv. (4.1 μL) of the mixture was taken and incubated with 1,4-butanedithiol (1 equiv.) at 37 °C for 30 m in 1 mL of water. 2 mg of myoglobin (1 equiv.) was dissolved in 2 mL of water and added to the reaction mixture. The reaction mixture was left to stir for 16 h at RT. The reaction mixture was purified by molecular weight cut-off and characterized by LC-MS to analyze the protein modification. Percent conversions to cyclized myoglobin were calculated based on the deconvolution spectra (38%).

## Procedure for labeling lysine in cell lysates

To block free thiol from forming disulfide linkage with the fluorophore, the free thiol on cell lysates were blocked using 100 μL solution of iodoacetamide (15 mM) in water for 1 h. The cell lysate was precipitated out using acetone and centrifuge at 1500 x g for 10 min at 4 °C. The supernatant was removed and the samples were ready for lysine labeling. To prepare for lysine labeling, furan (100 μL, 1.38 mmol, 1 equiv.) and sodium bicarbonate (115 mg, 1.38 mmol, 1 equiv.) were added to a solution of 12 mL ACN:H$_2$O (5:1). The reaction mixture was cooled to 0 °C and left to stir for 15 min. N-Bromosuccinimide (244 mg, 1.38 mmol, 1 equiv.) was dissolved in a solution of 12 mL ACN:H$_2$O (5:1) and added to the reaction mixture dropwise. Afterwards, the reaction mixture was left to stir for 10 min, and pyridine (222 μL, 2.76 mmol, 2 equiv.) was added to the reaction mixture. The reaction mixture was stirred for 4 h and used without further purification. From the pot, 1 mM of oxidized furan were incubated with 2 μL of 33 mM 1-(3',6'-dihydroxy-3-oxo-3H-spiro[isobenzofuran-1,9'-xanthen]-5-yl)-3-(2-mercaptoethyl) thiourea solution in 50 μL of water at 37 °C for 30 min. 200 μg of cysteine blocked cell lysate was dissolved in 200 μL of water and added to the reaction mixture. The reaction mixture was left to stir for 4 h at RT. Cell lysate was precipitated using cold acetone and analyzed using SDS-PAGE in-gel fluorescence analysis. Control experiments were performed by taking a stock solution from furan pot and adding it directly in a cell lysate sample and incubating it for 4 h followed by purification of protein by molecular weight and analysis by in-gel fluorescence analysis.

## Procedure for labeling cysteine in cell lysates

Furan (100 μL, 1.38 mmol, 1 equiv.) and sodium bicarbonate (115 mg, 1.38 mmol, 1 equiv.) were added to a solution of 12 mL ACN:H$_2$O (5:1). The reaction mixture was cooled to 0 °C and left to stir for 15 min. N-Bromosuccinimide (244 mg, 1.38 mmol, 1 equiv.) was dissolved in a solution of 12 mL ACN:H$_2$O (5:1) and added to the reaction mixture dropwise. Afterward, the reaction mixture was left to stir for 10 min, and pyridine (222 μL, 2.76 mmol, 2 equiv.) was added to the reaction mixture. The reaction mixture was stirred for 4 h and used without further purification. From the furan pot, 1 mM of oxidized furan was incubated with 100 μg of cell lysate in 100 μL of water and incubated at 37 °C for 10 min. 3 μL of 10 mM AZ680-amine dye in DMSO was added to the reaction mixture. The reaction mixture was left to stir for 4 h at RT. Cell lysate was precipitated using cold acetone and analyzed using SDS-PAGE in-gel fluorescence analysis. Control experiments were performed by taking a stock solution from a furan pot and adding it directly to a cell lysate sample and incubating it for 4 h followed by

purification of protein by molecular weight and analysis by in-gel fluorescence analysis

## Reporting summary

Further information on research design is available in the Nature Portfolio Reporting Summary linked to this article.

## Data availability

All data supporting the findings of this study are available within the Supplementary Information. These include the optimization of the reaction with oxidizing reagents, varying thiols and amines, the procedure of optimized reactions with small molecules, peptides, peptide macrocyclization, peptide stapling, protein modification, protein cyclization, dual modification of proteins, activity assay, cell lysate labeling and product characterization by NMR, HPLC, LC-MS, MS/MS and HRMS.

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

## Acknowledgements
This research was supported by NIH (Grant No. 1R35GM133719-01), NSF (Grant No. CHE–2108774) and American Cancer Society Research Scholar Grant (Grant No. RSG–22-025-01-CDP) to M.R.

## Author contributions
Y.W. and M.R. designed the project. Y.W. performed all the small molecule, peptide labeling, stapling and protein experiments and characterized the compounds by NMR and LC-MS. P.C. performed peptide cyclization experiments, modified furylalanine-containing peptides and characterized the compounds by HPLC and LC-MS. All authors analyzed the results. Y.W., P.C., and M.R. wrote the manuscript.

## Competing interests
The authors declare no competing interests.
