## [Peer Review File · Nature Communications]

Bioinspired One-pot Furan-Thiol-Amine Multicomponent Reaction for Making Heterocycles and its ApplicationsReviewers' Comments:

Reviewer #1:

Remarks to the Author:

The authors take inspiration from the cytochrome P450-mediated furan oxidation to generate dialdehyde as a building block to construct pyrrole while engaging amine and thiol. At first, they establish the methodology with relevant model substrates covering a range of electronic and steric attributes. Subsequently, they extend it to peptides, proteins, and cell lysates. In the process, they establish the potential of chemistry for targeting Lys and Cys in different sequences. However, it is a multi-step process rather than a multicomponent reaction in the true sense. The authors should rephrase the related segments to add clarity to the manuscript. Alternatively, the authors will have to demonstrate what happens when all the components are mixed together.

Peptide-related segment

1. Figure 4c outlines the stapling between (N-Ac) Lys and Cys. It would add value if the authors could examine the outcome of stapling with Cys in the presence of

(a) free N-terminus amine and one Lys,

(b) protected N-terminus amine with two Lys residues in the same peptide.

2. The authors should examine the combination of His/Lys and Arg/Lys in place of Cys/Lys. If the proposed mechanism in Figure 2b remains valid, these combinations should offer an alternative pathway to the bioconjugate. Since free Cys has a low occurrence in the proteome, the translation to cell lysate might not engage Cys/Lys combination exclusively. Hence, the potential competitors for Cys need to be explored.

Protein-related segment

3. The authors established the methodology with a set of proteins for Lys modification. However, this is the part that would draw significant criticism. There are already a few methods established for selective bioconjugation of Lys in proteins. Unfortunately, the current effort focuses on higher yields at the cost of selectivity. It is critical to check the extent of conversion while maintaining homogeneity or single-site labeling for all the examples. The authors should discuss whether it compares with this segment's pre-existing methods.

4. Another major concern is the need for more supporting data to validate the findings. For example, the conjugation is identified only in the case of myoglobin (K79). That, too, is a partial analysis as the bioconjugation results in labeling two sites. Identifying all the sites in heterogeneously labeled proteins will be challenging for the authors. However, the single site labeled bioconjugates recommended in the previous point would allow them to establish the kinetically preferred conjugation sites for all the examples. They should clearly specify the reoptimized conditions for homogeneous protein bioconjugation.

5. In Figure 6b, the conjugation sites are not validated by the MS data. It would add value but is not essential to establish the current claims. However, the free Cys-bearing protein such as BSA should also render the bioconjugate if thiol-fluorophore is replaced by amine-fluorophore (e.g., amine-rhodamine from Fig. 7c) while keeping other conditions constant. If this works, it supports labeling Cys-bearing proteome in the cell lysates, as claimed by the authors.

6. Figure 7b: It needs to be clarified what value the dual-labeling experiments add. The installation of the second probe utilized pre-established chemistry with residues that have no correlation with the reported chemistry. Besides, the conjugation site is not established in these cases.

7. Figure 7c: Methods or 7c footnotes do not specify the cells used to prepare the lysate. The authors should also specify how this lysate was prepared. The sample preparation conditions could have a substantial impact on the targetome. Importantly, there is no validation of the claims on Lys or Cys targeting in the proteome. It would help if authors perform the control experiments with Lys- and Cys-selective electrophiles and treat the sample again with their reagents to identify the bands that disappear. Also, the MS data would add substantial value to the validation.

8. Minor points: (a) page 2, lines 66-68, references 22-23: it is not relevant to the work done in this manuscript and can be avoided. (b) page 2, lines 76-77: What is 2 MCR? It can only be called MCR if there are at least three components. (c) Authors can check and edit the language: page 2, line 56:

"UGI" should be written as "Ugi," and "Passerine" should be written as "Passerini."

Overall, the manuscript outlines an interesting approach that could be potentially suitable for the journal if revised extensively.

Reviewer #2:

Remarks to the Author:

This work is about a new application of an old chemistry for peptide/protein modification under newly optimized conditions. Peterson's studies (refs 19, 20) have shown that furan and its analogs can be oxidized by P450 enzymes in vivo to form cis-2-Butene-1,4-dial (BDA). BDA can react with thiols and amines of peptides to form pyrrole-containing adducts including macrocyclic products. Recent work by Zheng (ref 21) reported a nice chemoproteomic application of this chemistry to profile adducted proteins derived from various furan compounds. In that work, Zheng showed that an epoxide reagent DMDO can convert furan to BDA as well. In the current work, Raj and coworkers demonstrated that the treatment of furan with NBS reagent in mixed organic solvents and water can also generate BDA. The resulting crude BDA solution can be used for reactions with amines and thiols to give the corresponding pyrrole-adduct with peptides and proteins. In comparison with the above previous reports, the current one demonstrated useful synthetic application of the BDA/thiol/amine reactions in peptide and protein modifications for the first time. This represents a meaningful advance of the old BDA chemistry. However, the method suffers from many drawbacks compared to a long list of well-established methods for peptide and protein modification involving amines or thiols. The BDA reagent must be prepared as a crude solution in organic solvents before use. The following addition operation needs to be carried out in two steps and requires long waiting time. The use of a significant amount of organic cosolvents would greatly hamper its application for more delicate biological systems. The reaction selectivity seems to be moderate, especially for longer peptide substrates. The pyrrole molecules are known to be quite reactive; their long-term stability might be an issue. Overall, this study reported synthetic applications of an old three-component reaction in peptide and protein modification under modified conditions. However, the novelty of such transformation is limited. The applications appear somewhat forced, and did not demonstrate much significant advantage over existing methods. Publication in more specialized journals such as bioconjugation chemistry or chemistry european journal would be more appropriate.

Reviewer #3:

Remarks to the Author:

Raj and coworkers describe a multicomponent reaction between a furan, a thiol, and an amine to yield modified pyrroles. The reaction is fast and efficient, and the authors explore its application in chemical biology. I agree with the authors that multicomponent reactions are an elegant methodology to generate complex structures, and that the inherent challenge of orthogonality hinders their use in bioconjugation. The authors draw inspiration from the CYP450 promoted formation of adducts between furan, GSH and cellular amines to develop this reaction that only occurs between these 3 partners upon oxidation of the furan, thus adding selectivity to the process. The broad scope shows compatibility with other functional groups and the methodology is extended to peptides and proteins. Importantly the authors perform the selective modification of myoglobin's tyrosine residues using diazonium followed by treatment with the furan and a fluorescent thiol, yielding a dual-modified protein.

The manuscript is well written, timely and should be of wide interest to the readership of Nature Communications. It is my belief that this thorough study on the multicomponent reaction of furans, thiols and amines as a bioconjugation tool should merit publication in Nature Communications, provided the authors' investigate/discuss the following issues:

1. When discussing the scope, the formation of 2h' as a minor scope is attributed to the formation of an oxazolidine intermediate and the reader is referred to Fig 3b. However the information is not explicit in the said figure, nor in the SI.
2. Some of the peptides used were prepared by SPPS. It would be useful to add the procedure and yield for the said peptides in the SI.
3. When discussing the chemoselective modification of myoglobin (Fig 5) the scheme depicts using ACN:H₂O (5:1). This mixture is incompatible with most proteins, so I believe that this is only for the oxidation step which is then diluted in the aqueous media. Please be more explicit on the final ratio of ACN:H₂O during the reaction.
4. On the same topic, I suggest performing circular dichroism experiments to confirm the retention of the protein tertiary structure.
5. A major challenge on targeting Lysine is the formation of heterogeneous conjugates, as encountered by the authors. Have any attempts on targeting the cysteine residues, using an external amine, been performed? Since the data points towards the thiol additions occurring before the condensation with amines, this approach should be viable and yield more homogeneous constructs.

Thank you for your consideration of this work. We thank the reviewers for a critical reading of the manuscript. We hope you will consider a revised version that addresses the key concerns of each reviewer.

We thank Reviewer #1 and #3 for the recommendation to publish after the revisions. We thank all the Reviewers #1 and #3 for the willingness to accept the manuscript once we answer all the comments. We have performed several experiments on peptides, proteins and a complex cell lysate and answered all the comments/concerns raised by reviewer #1, #2 and #3. As suggested, we have performed experiments and compared our method to existing ones for selective bioconjugation of lysine in proteins and data clearly showed that our method generated homogeneous proteins and is highly selective for lysine or cysteine as compared to the existing ones. This is the only method that can carry out dual labeling on lysine and cysteine in a selective manner using same chemistry by just changing different nucleophiles.

We have added all the new experiments and the corresponding data in the revised manuscript and supporting information as suggested by the reviewers. Our responses to the reviewers' comments are shown below.

Our responses to the reviewers' comments in details are shown below.

Reviewer #1 (Remarks to the Author):

The authors take inspiration from the cytochrome P450-mediated furan oxidation to generate dialdehyde as a building block to construct pyrrole while engaging amine and thiol. At first, they establish the methodology with relevant model substrates covering a range of electronic and steric attributes. Subsequently, they extend it to peptides, proteins, and cell lysates. In the process, they establish the potential of chemistry for targeting Lys and Cys in different sequences. However, it is a multi-step process rather than a multicomponent reaction in the true sense. The authors should rephrase the related segments to add clarity to the manuscript. Alternatively, the authors will have to demonstrate what happens when all the components are mixed.

We thank the reviewer for the comment but we wanted to point out that it is a multicomponent reaction because all the three reagents react to form a product and basically all or most of the atoms contributed to the newly formed product. We mixed all the three components together (BDA (synthesized in situ), thiol and amine) in a sequential manner. We have added "sequential manner" throughout the manuscript.

Peptide-related segment

1. Figure 4c outlines the stapling between (N-Ac) Lys and Cys. It would add value if the

authors could examine the outcome of stapling with Cys in the presence of
(a) free N-terminus amine and one Lys,
(b) protected N-terminus amine with two Lys residues in the same peptide.

We thank the reviewer for this insightful suggestion. As suggested, we synthesized two linear peptides, FMKNYC **1u**, containing one free N-terminus and one lysine along with cysteine and Ac-KFMKNYC **1v**, containing two N lysine along with cysteine. Both the linear peptides were subjected to the optimized peptide macrocyclization conditions. In both cases, two major peaks were observed in the crude HPLC spectra from these reactions, with each peak showing the mass of the cyclic pyrrole product from reaction of cysteine and amine with BDA. To determine the site of cyclization, the peptides were subjected to cleavage at methionine by CNBr. It was determined that each peak represented one of the two possible cyclization products, with no significant preference for one or the other as neither cycle is particularly strained. *We have included this data to Figure 4 of the manuscript and added the detailed experimental procedures and data into SI Figure 10.*

Changes in the manuscript:

Following new section and new data was added in the manuscript as Figure 4d:

Application of FuTine MCR in peptide macrocyclization and stapling.

To determine the preference for the formation of particular ring size during macrocyclization, we carried out FuTine reaction with linear peptides FMKNYC **1u** and Ac-KFMKNYC **1v**, containing either one-N-terminus and one lysine or two reactive lysine respectively. The results of cyclization on these peptides did not show any preference in regards to the ring sizes and both the macrocycles (**2u** and **2u'**) and (**2v** and **2v'**) are formed with almost similar conversions (Fig. 4d, Supplementary Fig. 10). The site of the macrocyclization was determined by cleaving the cyclic peptides (**2u** and **2u'**) and (**2v** and **2v'**) at Met using CNBr and resulting fragments were analyzed by MS (Supplementary Fig. 10). This study demonstrated the robustness of FuTine method for synthesis of both peptide and non-peptide macrocycles.

Fig. 4. FuTine MCR for chemoselective modification and cyclization of peptides. a Chemoselective modification of lysine or cysteine to form *N*-pyrrole on peptides without modification of any reactive amino acids. **b** Chemoselective modification of 2-furylalanine to form *N*-pyrrole on peptides. **c** Synthesis of peptide macrocycles of varying ring sizes between Cys and Lys side chains by the addition of oxidized furan. **d** Macrocyclization showed no preference for ring size, N-terminus, and reactive lysine residue **e** Stapling of peptides between two cysteine or two lysine residues on peptides by the addition of diamine or dithiol and oxidized furan.

Changes in the Supplementary Information:

Following changes were added to supplementary information Figure 10:

A solution of furan **1** (100 μ L, 1.4 mmol, 1 equiv.) and sodium bicarbonate (118 mg, 1.4 mmol, 1 equiv.) in 12 mL of ACN:H₂O (5:1) was incubated at 0 °C for 15 min. A 12 mL solution of NBS (244 mg, 1.4 mmol, 1 equiv.) in ACN:H₂O (5:1) was added to the mixture dropwise over 5 min at 0 °C and the reaction was stirred for 10 min. Pyridine (222 μ L, 2.8 mmol, 2 equiv.) was then added directly and the reaction mixture was allowed to stir at 0 °C for 4 h. This afforded the oxidized intermediate *cis*-2-butene-1,4-dial (BDA), which was not isolated. This reaction served as a stock solution of the reactive intermediate.

From a freshly prepared stock solution of prepared *cis*-2-butene-1,4-dial (BDA), a 114 μ L aliquot (7.5 μ mol, 1.2 equiv.) was added to a 6.0 mL solution of ACN:H₂O (5:1) containing 5 mg of peptide **1u** (6.2 μ mol, 1 equiv.). The reaction was allowed to stir at RT for 16 h. Following completion of the reaction, the solvent was removed using a centrifugal vacuum concentrator system. The product was re-dissolved in 600 μ L of ACN:H₂O (5:1) and analyzed by HPLC using method B to determine the percent conversion to the major products **2u** (31 %) and **2u'** (40 %), representing 71 % total conversion to cyclic pyrrole peptides. The peaks of these two major products were collected and lyophilized.

FMKNYC Linear Peptide 1u. LCMS m/z 804.3488 (calcd. $[M+H^+] = 804.3531$), m/z 402.6894 (calcd. $[(M+2H^+)/2] = 402.6802$). Purity: > 95 % (HPLC analysis at 220 nm). Retention time in HPLC: 9.726 min.

FMKNYC cyclic peptide 2u. LCMS m/z 852.3658 (calcd. $[M+H^+] = 852.3531$), m/z 426.6864 (calcd. $[(M+2H^+)/2] = 426.6765$). Retention time in HPLC: 16.395 min.

FMKNYC cyclic peptide 2u'. LCMS m/z 852.3665 (calcd. $[M+H^+] = 852.3531$), m/z 874.3476 (calcd. $[M+Na] = 874.3531$). Retention time in HPLC: 19.214 min.

Prep-HPLC trace of reaction mixture to generate 2u and 2u'

MS-trace of 1u

MS-trace of 2u

MS-trace of 2u'

CNBr cleavage to determine site of modifications

To determine the position of cyclization, cyanogen bromide (CNBr) was used to cleave the cyclic peptide at the methionine residue. To ~0.5 mg of products **2u** and **2u'** was added ~10 mg of CNBr in 500 μ L of 70 % aqueous formic acid (FA) in 2 mL glass vials which were capped and allowed to stir in the dark for 24 h, before careful evaporation of the acid and HRMS analysis. The results clearly showed the two major products by cyclization of the cysteine with either of the two amine residues.

N-terminal cleavage product of 2u. LCMS m/z 249.1268 (calcd. $[M+H]^+$ = 249.1234).

C-terminal cleavage product of 2u. LCMS m/z 590.2283 (calcd. $[M+H]^+$ = 590.2319).

Cleavage product of 2u'. LCMS m/z 822.3600 (calcd. $[M+H]^+$ = 822.3603), m/z 411.6951 (calcd. $[(M+2H^+)/2]$ = 411.6802).

MS trace of C & N terminus fragments of 2u after cleavage

MS trace of 2u' after cleavage

A solution of furan **1** (100 μ L, 1.4 mmol, 1 equiv.) and sodium bicarbonate (118 mg, 1.4 mmol, 1 equiv.) in 12 mL of ACN:H₂O (5:1) was incubated at 0 °C for 15 min. A 12 mL solution of NBS (244 mg, 1.4 mmol, 1 equiv.) in ACN:H₂O (5:1) was added to the mixture dropwise over 5 min at 0 °C and the reaction was stirred for 10 min. Pyridine (222 μ L, 2.8 mmol, 2 equiv.) was then added directly and the reaction mixture was allowed to stir at 0 °C for 4 h. This afforded the oxidized intermediate *cis*-2-butene-1,4-dial (BDA), which was not isolated. This reaction served as a stock solution of the reactive intermediate.

From a freshly prepared stock solution of prepared *cis*-2-butene-1,4-dial (BDA), a 94 μ L aliquot (6.2 μ mol, 1.2 equiv.) was added to a 6.0 mL solution of ACN:H₂O (5:1) containing 5 mg of peptide **1v** (5.1 μ mol, 1 equiv.). The reaction was allowed to stir at RT for 16 h. Following completion of the reaction, the solvent was removed using a centrifugal vacuum concentrator system. The product was re-dissolved in 600 μ L of ACN:H₂O (5:1) and analyzed by HPLC using method B to determine the percent conversion to the major products **2v** (38 %) and **2v'** (35 %), representing 73 % total conversion to cyclic pyrrole peptides. The peaks of these two major products were collected and lyophilized.

Ac-KFMKNYC linear peptide 1v. LCMS m/z 974.4534 (calcd. $[M+H]^+$ = 974.4587), m/z 487.7348 (calcd. $[(M+2H^+)/2]$ = 487.7330). Purity: > 95 % (HPLC absorbance at 220 nm). Retention time in HPLC: 13.3 min.

Ac-KFMKNYC cyclic peptide 2v. LCMS m/z 1022.4757 (calcd. $[M+H]^+$ = 1022.4587). Retention time in HPLC: 16.936 min.

Ac-KFMKNYC cyclic peptide 2v'. LCMS m/z 1022.4759 (calcd. $[M+H]^+$ = 1022.4587). Retention time in HPLC: 18.052 min.

HPLC trace of 1v

HPLC trace of reaction mixture to generate 2v and 2v'

MS trace of 1v

MS trace of 2v

MS trace of 2v'

To determine the position of cyclization, cyanogen bromide (CNBr) was used to cleave the cyclic peptide at the methionine residue. To ~0.5 mg of products **2v** and **2v'** was added ~10 mg of Cyanogen bromide (CNBr) in 500 μL of 70 % aqueous formic acid (FA) in 2 mL glass vials which were capped and allowed to stir in the dark for 24 h, before careful evaporation of the acid and HRMS analysis

N-terminal cleavage product of 2v. LCMS m/z 419.2385 (calcd. $[M+H]^+$ = 419.2289).

C-terminal cleavage product of 2v'. LCMS m/z 590.2361 (calcd. $[M+H]^+$ = 590.2319).

Cleavage product of 2v'. LCMS m/z 1008.4580 (calcd. $[M+H]^+$ = 1008.4535), m/z 504.7375 (calcd. $[(M+2H)^+]/2$ = 504.7268).

MS trace of C & N terminus fragments of 2v after cleavage

MS trace of 2v' after cleavage

2. The authors should examine the combination of His/Lys and Arg/Lys in place of Cys/Lys. If the proposed mechanism in Figure 2b remains valid, these combinations should offer an alternative pathway to the bioconjugate. Since free Cys has a low occurrence in the proteome, the translation to cell lysate might not engage Cys/Lys combination exclusively. Hence, the potential competitors for Cys need to be explored.

AS suggested, we have conducted the following experiments. Two linear peptides, Ac-KFR **1y** and Ac-KFH **1z**, were reacted under optimized conditions for the peptide macrocyclization. HPLC analysis confirmed only partial conversion to the 2H-pyrrol-2-one products **2y** and **2z** (for which a small molecule NMR was obtained for **4**, fig. 2c and SI Fig. 3) due to reaction of the lysine residue with BDA. We did not observe any reaction with neither arginine nor histidine. As demonstrated in the prior examples, no pyrrol-2-one product was obtained with lysine if both cysteine and lysine are present as shown in Fig. 2a. We have added the data for linear peptides **1y** and **1z** into the manuscript and supplementary information (SI Fig. 10).

Changes in the manuscript:

Following lines are added to the manuscript:

Similar attempts to synthesize macrocycles by reaction between His and Lys and the reaction between Arg and Lys did not work under the reaction conditions (Supplementary Fig. 10). These experiments further confirm the orthogonal nature of FuTine chemistry between Cys and Lys.

Changes in the supplementary information:

We have made the following changes to supplementary Figure 10.

A solution of furan **1** (100 μ L, 1.4 mmol, 1 equiv.) and sodium bicarbonate (118 mg, 1.4 mmol, 1 equiv.) in 12 mL of ACN:H₂O (5:1) was incubated at 0 °C for 15 min. A 12 mL solution of NBS (244 mg, 1.4 mmol, 1 equiv.) in ACN:H₂O (5:1) was added to the mixture dropwise over 5 min at 0 °C and the reaction was stirred for 10 min. Pyridine (222 μ L, 2.8 mmol, 2 equiv.) was then added directly and the reaction mixture was allowed to stir at 0 °C for 4 h. This afforded the oxidized intermediate cis-2-butene-1,4-dial (BDA), which was not isolated. This

reaction served as a stock solution of the reactive intermediate.

From a freshly prepared stock solution of prepared cis-2-butene-1,4-dial (BDA), a 38.8 μL aliquot (2.6 μmol , 1.2 equiv.) was added to a 1.0 mL solution of ACN:H₂O (5:1) containing 1 mg of peptide **1y** (2.1 μmol , 1 equiv.). The reaction was allowed to stir at RT for 16 h. Following completion of the reaction, the solvent was removed using a centrifugal vacuum concentrator system. The product was re-dissolved in 350 μL of ACN:H₂O (5:1) and analyzed by HPLC using method B to determine the percent conversion to the products **2y**, representing a total conversion of 49 % to the pyrrolin-2-one product and its hydrate.

Ac-KFH linear peptide 1y. LCMS m/z 472.2684 (calcd. $[M+H^+] = 472.2667$), m/z 263.6384 (calcd. $[(M+2H^+)/2] = 236.637$), m/z 943.5213 (calcd. $[2M+H^+] = 943.5261$). Purity: > 99 % (HPLC analysis at 220 nm). Retention time in HPLC: 3.711 min.

Ac-KFH linear peptide 1y. LCMS m/z 472.2733 (calcd. $[M+H^+] = 472.2667$, m/z 236.6403 (calcd. $[(M+2H^+)/2] = 236.6370$). Retention time in HPLC: 3.807 min.

Ac-KFH pyrrolin-2-one hydrate peptide 2y. LCMS m/z 538.2354 (calcd. $[M+H^+] = 538.2772$), m/z 556.2961 (calcd. $[M+H_2O+H^+] = 556.2878$). Retention time in HPLC: 9.193 min.

Ac-KFH pyrrolin-2-one peptide 2y. LCMS m/z 538.2850 (calcd. $[M+H^+] = 538.2772$), m/z 560.2668 (calcd. $[M+Na^+] = 560.2952$). Retention time in HPLC: 11.286 min.

HPLC trace of 1y

HPLC trace of reaction mixture to generate 2y

MS-trace 1y

MS-trace of reaction mixture peak at 3.807 min corresponding to unreacted 1y

MS-trace of reaction mixture peak at 9.193 min

MS-trace of reaction mixture peak at 11.286 min

A solution of furan **1** (100 μ L, 1.4 mmol, 1 equiv.) and sodium bicarbonate (118 mg, 1.4 mmol, 1 equiv.) in 12 mL of ACN:H₂O (5:1) was incubated at 0 °C for 15 min. A 12 mL solution of NBS (244 mg, 1.4 mmol, 1 equiv.) in ACN:H₂O (5:1) was added to the mixture dropwise over

5 min at 0 °C and the reaction was stirred for 10 min. Pyridine (222 μL , 2.8 mmol, 2 equiv.) was then added directly and the reaction mixture was allowed to stir at 0 °C for 4 h. This afforded the oxidized intermediate *cis*-2-butene-1,4-dial (BDA), which was not isolated. This reaction served as a stock solution of the reactive intermediate.

From a freshly prepared stock solution of prepared *cis*-2-butene-1,4-dial (BDA), a 37.4 μL aliquot (2.4 μmol , 1.2 equiv.) was added to a 1.0 mL solution of ACN:H₂O (5:1) containing 1 mg of peptide **1z** (2.0 μmol , 1 equiv.). The reaction was allowed to stir at RT for 16 h. Following completion of the reaction, the solvent was removed using a centrifugal vacuum concentrator system. The product was re-dissolved in 350 μL of ACN:H₂O (5:1) and analyzed by HPLC using method B to determine the percent conversion to the products **2z**, representing a total conversion of 39.6 % to the pyrrolin-2-one product and its hydrate.

Ac-KFR linear peptide 1z. LCMS m/z 491.3104 (calcd. $[M+H^+]$ = 491.3089), m/z 246.1604 (calcd. $[(M+2H^+)/2]$ = 246.1581). Purity: > 95 % (HPLC analysis at 220 nm). Retention time in HPLC: 4.177 min.

Ac-KFR linear peptide 1z after reaction. LCMS m/z 491.3159 (calcd. $[M+H^+]$ = 491.3089), m/z 246.1616 (calcd. $[(M+2H^+)/2]$ = 246.1581). Retention time in HPLC: 4.617 min.

Ac-KFR pyrrolin-2-one hydrate peptide 2z. LCMS m/z 557.3275 (calcd. $[M+H^+]$ = 557.3183), m/z 575.3378 (calcd. $[M+H_2O+H^+]$ = 575.3300). Retention time in HPLC: 9.702 min.

Ac-KFR pyrrolin-2-one peptide 2z. LCMS m/z 557.3270 (calcd. $[M+H^+]$ = 557.3183). Retention time in HPLC: 11.122 min.

HPLC trace of 1z

HPLC trace of reaction mixture to generate 2z

MS-trace 1z

MS-trace of reaction mixture peak at 4.617 min

MS-trace of reaction mixture peak at 9.702 min

MS-trace of reaction mixture peak at 11.122 min

3. The authors established the methodology with a set of proteins for Lys modification. However, this is the part that would draw significant criticism. There are already a few methods established for selective bioconjugation of Lys in proteins. Unfortunately, the current effort focuses on higher yields at the cost of selectivity. It is critical to check the extent of conversion while maintaining homogeneity or single-site labeling for all the examples. The authors should discuss whether it compares with this segment's pre-existing methods.

As suggested by the reviewer, we have added the condition for homogeneous labeling to the manuscript. We have made the following optimization on myoglobin: decreasing the reaction volume to 500 μ L (protein concentration in reaction: 236 μ M) and leaving the reaction for 8 hours instead of 16 hours. By doing so, we seen 91% conversion to single modify product. We have carried out this protocol on three other proteins (Cytochrome C, Lysozyme Egg, and aprotinin). In this case, we have seen single modification on all three of those protein by intact

mass analysis. However, when carry out MS/MS sequencing, we see two lysine K1 and K97 got modified simultaneously in the ratio of 3:1. Comparing with pre-existing method such as NHSester labeling of lysine, under the same conditions, we observed the formation of heterogenous products. It is also worth noting that a recent study using NHS ester showed reactivity with other amino acid residues such as arginine, serine, cysteine, tyrosine and threonine (please see the figure below). We have added the new section and data for homogenous labeling with our method in the revised manuscript (Figure 6) and heterogeneous labeling with NHS-ester in the revised manuscript and SI (SI, Fig. 16).

Ward, C. C.; Kleinman, J. I.; Nomura, D. K. NHS-Esters As Versatile Reactivity-Based Probes for Mapping Proteome-Wide Ligandable Hotspots. *ACS Chem. Biol.* 2017, 12, 6, 1478–1483. DOI: 10.1021/acscchembio.7b00125

Changes in the manuscript:

Application of FuTine MCR in generating homogenous proteins

In proteomics, homogeneous labeling of lysine residue remained challenging. Inspired by our initial screening on myoglobin (Fig. 5a), we attempted using 1.2 equiv. of thiol and BDA with a combination of shorter reaction time (8 h) and higher protein concentration in reaction to achieve homogeneous labeling of lysine (Fig. 6a, Supplementary Fig. 16). We carried out this optimized protocol on four different commercially available protein substrates (myoglobin, cytochrome C, lysozyme egg white, and aprotinin) and observed homogenous labeling in all the cases. We determined the sites of modification on proteins using MS/MS sequencing (Supplementary Fig. 16). In contrast, lysine modification with well-known NHS-ester led to the non-specific labeling to other amino acids such as Ser and Arg³⁰. In our hand, we also observed the formation of heterogenous mixture of products on myoglobin with NHS-ester under the reaction conditions (Supplementary Fig. 16). These results indicated that our

method is specific for labeling lysine and is amenable to generate homogenous products in high conversions.

Fig. 6. Homogeneous labeling of proteins using FuTine chemistry **a**. Optimized homogeneous labeling condition (1.2 equiv. of BDA & 1.2 equiv. of thioglycolic acid, 8 h, RT) of lysine residues on four different proteins. **b**. Duo labeling of cysteine and lysine residues on aprotinin with MS/MS analysis showed modification of K46, C30, & C38. We have also added a section in the manuscript to describe our homogeneous labeling methodology.

Changes in Supplementary Information

Following changes were adjusted to supplementary Figure 16:

XIX. Supplementary Fig. 16. Homogeneous labeling and determination the site of modification on homogeneous labeled proteins by LC-MS/MS

Homogeneous labeling of myoglobin

Furan **1** (100 μ L, 1.38 mmol) and sodium bicarbonate (115 mg, 1.38 mmol) were added to a solution of acetonitrile and water (10 mL and 2 mL, respectively). The reaction mixture was cooled to 0 °C and left to stir for 15 min. N-Bromosuccinimide (244 mg, 1.38 mmol) was dissolved in a solution of acetonitrile and water (10 mL and 2 mL, respectively) and added to the reaction mixture dropwise. Afterwards, the reaction mixture was left to stir for 10 min, and pyridine (222 μ L, 2.76 mmol) was added to the reaction mixture. The reaction mixture was stirred for 4 h and used without further purification. From the pot, 1.2 equiv. (2.5 μ L) of mixture was taken and incubated with thioglycolic acid (1.2 equiv.) at 37 °C for 30 min in 250 μ L of water. 2 mg of myoglobin (1 equiv.) was dissolved in 250 μ L of water and added to the reaction mixture (protein concentration in reaction: 236 μ M). The reaction mixture was left to stir for 8 h at RT. The reaction mixture was purified by molecular weight cut off and characterized by LCMS to analyze the protein modification. Percent conversions were calculated based on the deconvolution spectra.

MS spectrum of homogenous labeled myoglobin

Peptide fragment of homogeneous modified myoglobin

The site of modified myoglobin, obtained by treatment with 1.2 equiv. of oxidized furan and 1.2 equiv. of thioglycolic acid in 8 h was determined by Agilent Bioconfirm software after trypsin digestion using the SMART Digest™ Trypsin Kit by Thermo Scientific. The modification site was identified to be K79. The peptide fragment shown in the figures were

from AA residues 79-96 with the sequence KGHHEALKPLAQSHATK

Comparison with NHSester labeling of myoglobin

1.2 equiv. of 2,5-dioxopyrrolidin-1-yl benzoate was incubated with 2 mg of myoglobin (1 equiv.) dissolved in 0.1M NaHCO₃ solution. The reaction mixture was left to stir for 8 h at RT. The reaction mixture was purified by molecular weight cut off and characterized by LCMS to analyze the protein modification, which generated heterogeneous products. Percent conversions were calculated based on the deconvolution spectra.

MS spectrum of NHSester labeled myoglobin

Homogeneous labeling of cytochrome C

Furan **1** (100 μ L, 1.38 mmol) and sodium bicarbonate (115 mg, 1.38 mmol) were added to a solution of acetonitrile and water (10 mL and 2 mL, respectively). The reaction mixture was cooled to 0 °C and left to stir for 15 min. N-Bromosuccinimide (244 mg, 1.38 mmol) was dissolved in a solution of acetonitrile and water (10 mL and 2 mL, respectively) and added to the reaction mixture dropwise. Afterwards, the reaction mixture was left to stir for 10 min, and pyridine (222 μ L, 2.76 mmol) was added to the reaction mixture. The reaction mixture was stirred for 4 h and used without further purification. From the pot, 1.2 equiv. (3.3 μ L) of mixture was taken and incubated with thioglycolic acid (1.2 equiv.) at 37 °C for 30 min in 250 μ L of water. 2 mg of cytochrome C (1 equiv.) was dissolved in 250 μ L of water and added to the reaction mixture (protein concentration in reaction mixture: 324 μ M). The reaction mixture was left to stir for 8 h at RT. The reaction mixture was purified by molecular weight cut off and characterized by LCMS to analyze the protein modification. Percent conversions were calculated based on the deconvolution spectra.

MS spectrum of homogenous labeled cytochrome C

Peptide fragment of homogeneous modified Cytochrome C

The site of modified on Cytochrome C, obtained by treatment with 1.2 equiv. of oxidized furan and 1.2 equiv. of thioglycolic acid in 8 h was determined by Agilent Bioconfirm software after trypsin digestion using the SMART Digest™ Trypsin Kit by Thermo Scientific. The modification site was identified to be K8. The peptide fragment shown in the figures were from AA residues 8-13 with the sequence KIFVQK

A: Chain A

Homogeneous labeling of lysozyme egg white

Furan **1** (100 μ L, 1.38 mmol) and sodium bicarbonate (115 mg, 1.38 mmol) were added to a solution of acetonitrile and water (10 mL and 2 mL, respectively). The reaction mixture was cooled to 0 °C and left to stir for 15 min. N-Bromosuccinimide (244 mg, 1.38 mmol) was dissolved in a solution of acetonitrile and water (10 mL and 2 mL, respectively) and added to the reaction mixture dropwise. Afterwards, the reaction mixture was left to stir for 10 min, and pyridine (222 μ L, 2.76 mmol) was added to the reaction mixture. The reaction mixture was stirred for 4 h and used without further purification. From the pot, 1.2 equiv. (2.9 μ L) of mixture was taken and incubated with thioglycolic acid (1.2 equiv.) at 37 °C for 30 min in 250 μ L of water. 2 mg of lysozyme egg white (1 equiv.) was dissolved in 250 μ L of water and added to the reaction mixture (protein concentration in reaction mixture: 280 μ M). The reaction mixture was left to stir for 8 h at RT. The reaction mixture was purified by molecular weight cut off and characterized by LCMS to analyze the protein modification. Percent conversions were calculated based on the deconvolution spectra.

MS spectrum of homogenous labeled lysozyme egg white

Peptide fragment of homogeneous modified lysozyme egg white

The site of modified on Cytochrome C, obtained by treatment with 1.2 equiv. of oxidized furan and 1.2 equiv. of thioglycolic acid in 8 h was determined by Agilent Bioconfirm software after trypsin digestion using the SMART Digest™ Trypsin Kit by Thermo Scientific. The modification site was identified to be K1 and K97 (3:1). The peptide fragment shown in the figures were from AA residues 1-5 with the sequence KVFGR and AA residues KIVSDGNGMNAW

A: Chain A

Exact Mass: 745.36

Exact Mass: 1430.60

Homogeneous labeling of aprotinin

Furan **1** (100 μ L, 1.38 mmol) and sodium bicarbonate (115 mg, 1.38 mmol) were added to a solution of acetonitrile and water (10 mL and 2 mL, respectively). The reaction mixture was cooled to 0 °C and left to stir for 15 min. N-Bromosuccinimide (244 mg, 1.38 mmol) was dissolved in a solution of acetonitrile and water (10 mL and 2 mL, respectively) and added to the reaction mixture dropwise. Afterwards, the reaction mixture was left to stir for 10 min, and pyridine (222 μ L, 2.76 mmol) was added to the reaction mixture. The reaction mixture was stirred for 4 h and used without further purification. From the pot, 1.2 equiv. (6.4 μ L) of mixture was taken and incubated with thioglycolic acid (1.2 equiv.) at 37 °C for 30 min in 250 μ L of water. 2 mg of lysozyme egg white (1 equiv.) was dissolved in 250 μ L of water and added to the reaction mixture (protein concentration in reaction mixture: 614 μ M). The reaction mixture was left to stir for 8 h at RT. The reaction mixture was purified by molecular weight cut off and characterized by LCMS to analyze the protein modification. Percent conversions were calculated based on the deconvolution spectra.

MS spectrum of homogenous labeled aprotinin

Peptide fragment of homogeneous modified Aprotinin

The site of modified on Aprotinin, obtained by treatment with 1.2 equiv. of oxidized furan and 1.2 equiv. of thioglycolic acid in 8 h was determined by Agilent Bioconfirm software after trypsin digestion using the SMART Digest™ Trypsin Kit by Thermo Scientific. The modification site was identified to be K46. The peptide fragment shown in the figures were from AA residues 46-53 with the sequence KSAEDCMR

A: Chain A

1 **RPDFCLEPPYTGPKAR**IIIRYFYN**AKAGLCQTFVYGGCR**AKRRNNFK**SAEDCMRTC**GGGA

58

Exact Mass: 1135.41

Alkylation (iodoacetamide) comes from protein digestion

4. Another major concern is the need for more supporting data to validate the findings. For example, the conjugation is identified only in the case of myoglobin (K79). That, too, is a partial analysis as the bioconjugation results in labeling two sites. Identifying all the sites in

heterogeneously labeled proteins will be challenging for the authors. However, the single site labeled bioconjugates recommended in the previous point would allow them to establish the kinetically preferred conjugation sites for all the examples. They should clearly specify the reoptimized conditions for homogeneous protein bioconjugation.

We thank the reviewer for a great suggestion. We analyzed the MS/MS data again on modified myoglobin and observed that the 2nd modification on myoglobin occurred at K63 position but at a significantly lower abundance as compared to K79. We have determined the ratio in terms of abundance and included the MS/MS fragment of K63 in Figure 5 and full MS/MS data to the SI (SI, Fig. 13)

Changes in Manuscript

Next, we carried out MS/MS analysis on a myoglobin sample that was treated with 1 equiv. of oxidized furan and thioglycolic acid and determined the sites of modification on myoglobin to be K79 and K63 in the ratio of 3:1 (Fig. 5b, Supplementary Fig. 13).

Fig. 5. One-pot FuTine MCR for chemoselective modification of proteins. **a** Optimization of the selective modification of lysine on a protein myoglobin by varying equivalents of both oxidized furan (1-15 equiv.) and thiol (1-15 equiv.) reagents. Full conversion was observed by using 5 equiv. of thiol and furan indicating high robustness and efficiency of the thiol-amine reaction. **b** MS/MS analysis of modified myoglobin (treated with 1 equiv. of thiol and furan) showed the modification of K79 and K63 (3:1). **c** pyrrole-modified myoglobin (pyr-Myo) under optimized condition (treatment with 5 equiv. of thiol and furan) demonstrate similar ability to oxidize o-phenylenediamine comparing with unmodified myoglobin. This data supports the hypothesis that the 3D structure of the myoglobin remained intact after the modification which was further verified by circular dichroism analysis.

Changes in Supplementary Information

We have added the following information to Supplementary figure 13.

Peptide fragments of modified myoglobin under 16 hours of incubation. The site of modified myoglobin, obtained by treatment with 1.2 equiv. of oxidized furan and 1.2 equiv. of thioglycolic acid was determined by Agilent Bioconfirm software after trypsin digestion using the SMART Digest™ Trypsin Kit by Thermo Scientific. The modification site was identified to be K79 and K63 (3:1). The peptide fragment shown in the figures were from AA residues 79-96 with the sequence KGHHEAELKPLAQSHATK and peptide fragment from AA residues 63-77 with the sequence KHGTVVLTALGGILK.

Exact Mass: 1645.92

5. In Figure 6b, the conjugation sites are not validated by the MS data. It would add value but is not essential to establish the current claims. However, the free Cys-bearing protein such as BSA should also render the bioconjugate if thiol-fluorophore is replaced by amine-fluorophore (e.g., amine-rhodamine from Fig. 7c) while keeping other conditions constant. If this works, it supports labeling Cys-bearing proteome in the cell lysates, as claimed by the authors.

As suggested, we carried out a cystine labeling on three proteins BSA, transferrin and creatine kinase using amine-rhodamine as fluorophore and carried out SDS-PAGE gel analysis. *The new data for labeling cysteine in three proteins is added in Fig. 7b as below in the revised manuscript. We have added those gel data to the manuscript in figure 7b and adjusted the caption accordingly.*

For protein, Aprotinin, we carried out dual labeling of both lysine and cysteine using FuTine chemistry. This is the first time that the same chemistry was used for labeling two amino acids in such a precise manner. We also carried out MS/MS sequencing and found that K 46, and C30 and C38 got modified with our chemistry. *The dual labeling data using FuTine chemistry is added as Fig. 6b as below. For detailed procedure and data (MS/MS), we added the data in SI (Fig. 17 and Fig. 19).*

Changes in the manuscript:

Furthermore, we have also carried out duo labeling of cysteine and lysine separately using FuTine chemistry on aprotinin (Fig. 6b, Supplementary Fig. 17). We first homogeneously labeled K46 residue on aprotinin followed by cleaving the disulfide bond using dithiothreitol. We then carried out FuTine labeling of cysteine residues using 5 equiv. of BDA and amine, giving a duo labeled aprotinin on cysteine (C30 & C38) and lysine (K46) as confirmed by MS/MS sequencing. There is no current methodology in literature which can target both cysteine and lysine residue together on proteins using same chemistry.

We next utilized FuTine chemistry for the selective fluorescent labeling of cysteine on the same three native proteins, BSA, creatine kinase and transferrin. We incubated the proteins with dithiothreitol for 1 h to cleave the disulfide bonds and incubated the sample with oxidized furan and amine-rhodamine for 16 h at room temperature followed by protein precipitation and analysis by SDS-PAGE using in-gel fluorescence. The results clearly showed the labeling of BSA, creatine kinase and transferrin with fluorophores in the presence of all three reaction components (lanes 4, Fig. 7b, Supplementary Fig. 19). No fluorophore labeling was observed in control experiments in the absence of even one component (lanes 1-3, Fig. 7b).

Fig. 6. Homogenous labeling of proteins using FuTine chemistry **b** Duo labeling of cysteine and lysine residues on aprotinin with MS/MS analysis showed modification of K46, C30, & C38.

Fig. 7. Application of FuTine MCR for chemoselective modification of diverse proteins. **a** Chemoselective modification of lysine on proteins of varying sizes (5-79 KDa) and 3D-structures with > 99% conversion in all the cases under optimized conditions analyzed by MS. **b** Fluorophore labeling of lysine and cysteine in native proteins and their analysis by in-gel fluorescence.

Changes in the Supplementary Information:

The following information has been added to supplementary figure 17:

XXI. Supplementary Fig. 17. Duo modification of cysteine and lysine residues on aprotinin

Furan (100 μ L, 1.38 mmol) and sodium bicarbonate (115 mg, 1.38 mmol) were added to a solution of acetonitrile and water (10 mL and 2 mL, respectively). The reaction mixture was cooled to 0 °C and left to stir for 15 min. N-Bromosuccinimide (244mg, 1.38 mmol) was dissolved in a solution of acetonitrile and water (10 mL and 2 mL, respectively) and added to the reaction mixture dropwise. Afterwards, the reaction mixture was left to stir for 10 min, and pyridine (222 μ L, 2.76 mmol) was added to the reaction mixture. The reaction mixture was stirred for 4 h and used without further purification.

6.5 mM of dithiothreitol in water was prepared and K46 labeled aprotinin were incubated in the DTT solution (1 mL) at room temperature prior to FuTine labeling.

From the furan pot, 5 equiv. of mixture was taken and incubated with the protein samples for 15 minutes. Propargylamine (5 equiv.) was added to the reaction mixture and was left to stir for 16 h at RT. The reaction mixture was purified by molecular weight cut off and characterized by MS/MS sequencing using the SMART Digest™ Trypsin Kit by Thermo Scientific. The modification site was identified by Agilent Bioconfirm software to be C30 and C38 (1:1). The peptide fragment shown in the figures were from AA residues 22-39 with the sequence FYNAKAGLCQTFVYGGCR

We have added the detailed experimental protocol into supplementary figure 19

Fluorescent labeling of cysteine residues on BSA, creatine kinase and Transferrin

Furan (100 μ L, 1.38 mmol) and sodium bicarbonate (115 mg, 1.38 mmol) were added to a solution of acetonitrile and water (10 mL and 2 mL, respectively). The reaction mixture was cooled to 0 °C and left to stir for 15 min. N-Bromosuccinimide (244mg, 1.38 mmol) was dissolved in a solution of acetonitrile and water (10 mL and 2 mL, respectively) and added to the reaction mixture dropwise. Afterwards, the reaction mixture was left to stir for 10 min, and pyridine (222 μ L, 2.76 mmol) was added to the reaction mixture. The reaction mixture was stirred for 4 h and used without further purification.

6.5 mM of dithiothreitol in water was prepared and protein samples were incubated in the DTT solution (1 mL) at room temperature prior to FuTine labeling.

From the furan pot, 5 equiv. of mixture was taken and incubated with the protein samples (2 mg of BSA, creatine kinase or transferrin) for 15 minutes. AZ680 amine dye (5 equiv.) was added to the reaction mixture and was left to stir for 16 h at RT. The reaction mixture was purified by molecular weight cut off and characterized by in-gel fluorescence to analyze the protein modification. Control experiments were performed by taking a stock solution from furan pot and adding it directly to 2 mg of the protein (BSA, transferrin or Creatine kinase) and incubating it for 16 h followed by purification of protein by molecular weight and analysis by in-gel fluorescence analysis.

Full gel image of Creatine Kinase

Full gel image of Transferrin Human

Full gel image of BSA

6. Figure 7b: It needs to be clarified what value the dual-labeling experiments add. The installation of the second probe utilized pre-established chemistry with residues that have no correlation with the reported chemistry. Besides, the conjugation site is not established in these cases.

The purpose of dual labeling is to show that FuTine chemistry is compatible with other existing methods.

We also performed new experiments and showed that same chemistry can be used for dual labeling of cysteine and lysine just by changing the nucleophiles. This is the first report where same chemistry is used for labeling two residues by changing the nucleophiles. The data has been added in fig. 6b and SI Fig. 17 (as shown in the point above).

7. Figure 7c: Methods or 7c footnotes do not specify the cells used to prepare the lysate. The authors should also specify how this lysate was prepared. The sample preparation conditions could have a substantial impact on the targetome. Importantly, there is no validation of the claims on Lys or Cys targeting in the proteome. It would help if authors perform the control experiments with Lys- and Cys-selective electrophiles and treat the sample again with their reagents to identify the bands that disappear. Also, the MS data would add substantial value to the validation.

As suggested, we have added the cell lysate preparation procedures in SI Fig. 22. We have taken the advice from the reviewer and performed a competition inhibition assay to evaluate the selectivity. Two well-known reagents were selected: iodoacetamide for cysteine and NHSester for lysine. For cysteine labeling, we have incubated the cell lysate with various concentrations of iodoacetamide (0 μ M, 500 μ M, 1 mM) followed by FuTine cysteine labeling. Comparing with no treatment with iodoacetamide (lane 1, SI Figure 22), we have observed dose-dependence difference in the fluorescent intensity from FuTine chemistry. For lysine labeling, we treated the cell lysate with various concentrations of NHSester analog (0 μ M, 1 mM, 5mM) followed by FuTine lysine labeling. Comparing with no treatment with NHSester (lane 1, SI Figure 22), we have observed dose-dependence difference in fluorescent intensity from FuTine chemistry. Furthermore, we have confirmed the selective labeling of Lys or Cys targeting in the proteome with intact mass analysis, MS/MS sequencing, and fluorophore labeling with more than ten types of proteins.

Our data clearly showed that at the high concentrations of competitive inhibitor, we observed low labeling with FuTine chemistry, which further confirmed our claim that FuTine Chemistry is selective and can be used for profiling both cysteine and lysine by using different nucleophiles. This new data has been added in SI Figure. 22.

Changes in the manuscript:

The following changes were made in the manuscript.

To further confirm our claims on targeting either lysine or cysteine in the proteome by using FuTine chemistry. We have carried out a competition inhibition assay in both cases. We used iodoacetamide, a well-known cysteine probe, in various concentration to block free cysteine residues on cell lysate. We then incubated the treated cell lysate sample with oxidized furan and amine-rhodamine. The proteins were precipitated for analysis by in-gel florescence, which clearly showed differences in florescence intensity between samples that were treated with iodoacetamide and samples that weren't in dose-dependent manner (Supplementary Fig. 22). For targeting lysine, we incubated cell lysate with different concentrations of NHS-ester

followed by treatment with oxidized furan and thiol-FITC. As expected, we observed differences in fluorescence intensity between samples that were treated with NHS-ester and samples that weren't in a dose dependent manner (Supplementary Fig. 22). This assay confirmed the selectivity of FuTine chemistry for lysine and cysteine and the ability of FuTine chemistry for carrying out chemoproteomic profiling of cysteine and lysine residues in a complex cell lysate.

Changes in the Supplementary Information:

The following changes were added in supplementary figure 22

Cell lysate preparation procedure. LnCap cells were removed from the incubator and the media was aspirated immediately using vacuum. Cells were washed with ice cold PBS and aspirated again. Cells were scraped out of the dish on ice and transferred to 1.5 mL centrifuge tubes. Tubes were centrifuged for 5 minutes at 2000 rpm at 4 °C to pellet the cells and the excess PBS was removed and centrifuged tubes were kept on ice. Cell pellets were resuspended in RIPA buffer and incubated on ice for 10 minutes. The tubes were centrifuged at 13000 rpm for 10 minutes at 4 °C. The supernatants was removed to the final storage tubes and stored at -20 °C.

Competition inhibition assay for lysine labeling.

To block free thiol from forming disulfide linkage with the fluorophore, the free thiol on cell lysates were blocked using 100 μ L solution of iodoacetamide (15 mM) in water for 1 hour. The reaction. The cell lysates were precipitated out using acetone and centrifuge at 5000 rpm for 10 min at 4 °C. The supernatant was removed, and the sample was incubated with a various concentrations of 2,5-dioxopyrrolidin-1-yl benzoate (NHS-ester) in 0.1 M NaHCO₃ for 4 h at room temperature. The cell lysates were precipitated out using acetone and centrifuge at 5000 rpm for 10 minutes at 4 °C. To prepare for lysine labeling, furan (100 μ L, 1.38 mmol) and sodium bicarbonate (115 mg, 1.38 mmol) were added to a solution of acetonitrile and water (10 mL and 2 mL, respectively). The reaction mixture was cooled to 0 °C and left to stir for 15 min. N-Bromosuccinimide (244 mg, 1.38 mmol) was dissolved in a solution of acetonitrile and water (10 mL and 2 mL, respectively) and added to the reaction mixture dropwise. Afterwards, the reaction mixture was left to stir for 10 min, and pyridine (222 μ L, 2.76 mmol) was added to the reaction mixture. The reaction mixture was stirred for 4 h and used without further purification. From the pot, 1 mM of oxidized furan were incubated with 2 μ L of 33 mM (HS-FITC) 1-(3',6'-dihydroxy-3-oxo-3H-spiro[isobenzofuran-1,9'-xanthen]-5-yl)-3-(2-mercaptoethyl) thiourea solution in 50 μ L of water at 37 °C for 30 min. ~100 μ g of cell lysates treated with

varying concentrations of NHS-ester were dissolved in 200 μL of water and added to the reaction mixtures. The reaction mixtures were left to stir for 4 h at RT. Cell lysates were precipitated using cold acetone and analyzed using SDS-PAGE in-gel fluorescence analysis. The decrease in fluorescent intensity was observed with increasing concentrations of NHSester. Lane 1: No NHSester, Lane 2: 1 mM NHSester, Lane 3: 5 mM NHSester.

Full gel imaging of lysine competition assay

Competition inhibition assay for cysteine labeling

The cell lysate was incubated with varying concentrations of iodoacetamide (50 and 100 μM) for 1 hour. The cell lysates were precipitated out using acetone and centrifuge at 5000 rpm for 10 min at 4 °C. To prepare for cysteine labeling by FuTine chemistry, furan (100 μL , 1.38 mmol) and sodium bicarbonate (115 mg, 1.38 mmol) were added to a solution of acetonitrile and water (10 mL and 2 mL, respectively). The reaction mixture was cooled to 0 °C and left to stir for 15 min. N-Bromosuccinimide (244 mg, 1.38 mmol) was dissolved in a solution of acetonitrile and water (10 mL and 2 mL, respectively) and added to the reaction mixture dropwise. Afterwards, the reaction mixture was left to stir for 10 min, and pyridine (222 μL , 2.76 mmol) was added to the reaction mixture. The reaction mixture was stirred for 4 h and used without further purification. From the furan pot, 1 mM of oxidized furan were incubated with 100 μg of cell lysates treated previously with varying concentrations of iodoacetamide in 100 μL of water and incubated at 37 °C for 10 min. 2 μL of 10 mM AZ680 amine dye in DMSO were added to the reaction mixtures. The reaction mixtures were left to stir for 4 h at room temperature. Cell lysates were precipitated using cold acetone and analyzed using

SDS-PAGE in-gel fluorescence analysis. Control experiments were performed by taking a stock solution from furan pot and adding it directly in a cell lysate sample and incubating it for 4 h followed by purification of protein by molecular weight and analysis by in-gel fluorescence analysis. The decrease in fluorescent intensity by FuTine chemistry was observed with increase in the concentrations of NHSester. Lane 1: No NHSester, Lane 2: 500 μ M iodoacetamide, Lane 3: 1 mM iodoacetamide.

Full gel imaging of cysteine competition assay

8. Minor points: (a) page 2, lines 66-68, references 22-23: it is not relevant to the work done in this manuscript and can be avoided. (b) page 2, lines 76-77: What is 2 MCR? It can only be called MCR if there are at least three components. (c) Authors can check and edit the language: page 2, line 56: “UGI” should be written as “Ugi,” and “Passerine” should be written as “Passerini.”

The authors would like to thank the reviewer for pointing out those minor corrections in the manuscript. All of the suggestions were taken into considerations and adjustments were made in the manuscript.

Overall, the manuscript outlines an interesting approach that could be potentially suitable for the journal if revised extensively.

The author would like to thank the reviewer for reading the manuscript carefully and thoughtfully. We had revised the manuscript extensively and we thank reviewer #1 for encouraging comments and recommendation to publish after revisions.

Reviewer #2 (Remarks to the Author):

This work is about a new application of an old chemistry for peptide/protein modification under newly optimized conditions. Peterson’s studies (refs 19, 20) have shown that furan and its analogs can be oxidized by P450 enzymes in vivo to form cis-2-Butene-1,4-dial (BDA).

BDA can react with thiols and amines of peptides to form pyrrole-containing adducts including macrocyclic products. Recent work by Zheng (ref 21) reported a nice chemoproteomic application of this chemistry to profile adducted proteins derived from various furan compounds. In that work, Zheng showed that an epoxide reagent DMDO can convert furan to BDA as well. In the current work, Raj and coworkers demonstrated that the treatment of furan with NBS reagent in mixed organic solvents and water can also generate BDA. The resulting crude BDA solution can be used for reactions with amines and thiols to give the corresponding pyrrole-adduct with peptides and proteins. In comparison with the above previous reports, the current one demonstrated useful synthetic application of the BDA/thiol/amine reactions in peptide and protein modifications for the first time. This represents a meaningful advance of the old BDA chemistry. However, the method suffers from many drawbacks compared to a long list of well-established methods for peptide and protein modification involving amines or thiols.

We disagree with reviewer's comment in the following way. One of the most well-established method for labeling lysine residues, NHSester is not chemoselective (please see our answer to reviewer #1 Question 3).

The BDA reagent must be prepared as a crude solution in organic solvents before use. The following addition operation needs to be carried out in two steps and requires long waiting time. The use of a significant amount of organic cosolvents would greatly hamper its application for more delicate biological systems.

We thank the reviewer for point this out. We believed that there was a misunderstanding in our figure 5 and 6 (which we have fixed). It is true that BDA reagent used were prepared in 5:1 ACN: H₂O as a stock solution. However, only a negligible amount of ACN were used in the protein labeling vials. For instance, in our optimized conditions of labeling ~10 μL of BDA stock solution in 5:1 ACN: H₂O was added to 3 mL of myoglobin dissolved in pure water. Through some calculations, the final concentration of ACN in water during protein labeling was just 0.27%. The detailed procedures for protein labeling were in the SI (Supplementary Fig. 13).

The reaction selectivity seems to be moderate, especially for longer peptide substrates. The pyrrole molecules are known to be quite reactive; their long-term stability might be an issue.

We have observed >75% conversion with all our peptides (10 examples) and proteins (more than 10 examples) regardless of the size and 3D-structures as shown in Fig. 4 to Fig. 7. We have screened through a variety of different reactive functional groups on our chemoselective peptides and proteins (Fig. 4a and Fig. 5 to Fig.7) and found no selectivity issues with any other residue under the reaction conditions.

We would like to point the reviewer's attention to the studies we carried out on the modified myoglobin to determine the stability of pyrrole by leaving a modified myoglobin in various pH (3-11) for 24 h. The results showed no modification or degradation. The data is reported in Supplementary Fig. 14.

Overall, this study reported synthetic applications of an old three-component reaction in peptide and protein modification under modified conditions. However, the novelty of such transformation is limited. The applications appear somewhat forced, and did not demonstrate much significant advantage over existing methods.

We have demonstrated a wide range of applications of FuTine method which are as follows: In Fig 4, we have demonstrated the use of FuTine chemistry for (i) chemoselective single amino acid residue labeling, (ii) peptide macrocyclization, and (iii) peptide stapling.

In Fig 5, Fig 6, and Fig 7, we have demonstrated FuTine chemistry (i) for the selective labeling of proteins independent of size and 3D structures, (ii) duo labeling of Cys and Lys residues based on the nucleophile introduced, (iii) synthesis of homogenous proteins which is in contrast to NHS-ester for labeling lysine and (iv) a potential useful tool for activity-based protein profiling of lysine and cysteine in a complex cell lysate.

Reviewer #3 (Remarks to the Author):

Raj and coworkers describe a multicomponent reaction between a furan, a thiol, and an amine to yield modified pyrroles. The reaction is fast and efficient, and the authors explore its application in chemical biology. I agree with the authors that multicomponent reactions are an elegant methodology to generate complex structures, and that the inherent challenge of orthogonality hinders their use in bioconjugation. The authors draw inspiration from the CYP450 promoted formation of adducts between furan, GSH and cellular amines to develop this reaction that only occurs between these 3 partners upon oxidation of the furan, thus adding selectivity to the process. The broad scope shows compatibility with other functional groups and the methodology is extended to peptides and proteins. Importantly the authors perform the selective modification of myoglobin's tyrosine residues using diazonium followed by treatment with the furan and a fluorescent thiol, yielding a dual-modified protein.

The manuscript is well written, timely and should be of wide interest to the readership of Nature Communications. It is my belief that this thorough study on the multicomponent reaction of furans, thiols and amines as a bioconjugation tool should merit publication in Nature Communications, provided the authors' investigate/discuss the following issues:

The author would like to thank the reviewer for the encouraging comments and willing to accept it after minor corrections.

1. When discussing the scope, the formation of 2h' as a minor scope is attributed to the formation of an oxazolidine intermediate and the reader is referred to Fig 3b. However the information is not explicit in the said figure, nor in the SI.

Changes in the Supplementary Information:

We thank the reviewer for pointing this out to this. The mechanism for 2h' is added to the SI Fig 4 and corresponding content in the manuscript was changed.

2. Some of the peptides used were prepared by SPPS. It would be useful to add the procedure and yield for the said peptides in the SI.

The authors would like to thank the reviewer for this suggestion. Procedure for SPPS were included in the supplementary information section V. The yields and starting material peptides were not reported because the authors did not purify the whole peptide but the sufficient amount to conduct the precedent study.

Changes in the Supplementary Information:

Supplementary information section V:

Fmoc Solid-Phase Peptide Synthesis (Fmoc-SPPS).¹ Peptides were synthesized using standard protocols. Peptides were synthesized manually on a 0.25 mmol scale using Rink amide resin. Resin was swollen with DCM for 30 min at RT. Fmoc was deprotected using 20 % piperidine–DMF for 15 min to obtain a deprotected resin. Fmoc protected amino acid (1.25 mmol, 5 equiv.) was coupled using HOBT (1.25 mmol, 5 equiv.) and DIC (1.25 mmol, 5 equiv.) in DMF for 15 min at RT. Fmoc-protected amino acids (0.75 mmol, 3 equiv.) were sequentially coupled on the resin using HOBT (1.25 mmol, 5 equiv.) and DIC (1.25 mmol, 5 equiv.) in DMF for 15 min at RT. Peptides were cleaved from the resin using 4 mL of a cocktail consisting of 95:2.5:2.5 trifluoroacetic acid : water : triisopropylsilane (TIS) for 2 h. The resin was removed by filtration and the resulting solution was concentrated. Peptides were precipitated and centrifugated with cold diethyl ether (3 x 10 mL) to obtain the crude product. Crude peptides were dissolved in ACN:H₂O and purified by HPLC

3. When discussing the chemoselective modification of myoglobin (Fig 5) the scheme depicts using ACN:H₂O (5:1). This mixture is incompatible with most proteins, so I believe that this is only for the oxidation step which is then diluted in the aqueous media. Please be more explicit on the final ratio of ACN:H₂O during the reaction.

Thank you for pointing this out. The reviewer is absolutely correct, ACN:H₂O (5:1) was used only for the oxidation step for making a stock solution. The protein modification step was done in 3 mL of water and 10 μL of stock solution of ACN:H₂O (5:1) so overall the protein modification reaction was done in >99% water which is compatible with proteins. Adjustments

were made on the figures and manuscript, including clearly stating the procedures in the SI as well as in the manuscript.

4. On the same topic, I suggest performing circular dichroism experiments to confirm the retention of the protein tertiary structure.

The authors would like to thank the reviewer for this suggestion. A circular dichroism experiment were performed on modified myoglobin using optimized conditions. The protein was then analyzed and compared with wild type myoglobin. *This piece of data was added to the manuscript Fig. 5c and SI Fig. 15.*

Changes in the Manuscript:

We also carried out the circular dichroism CD studies with both modified and unmodified myoglobin and observed no change in the secondary structure after the modification (Fig. 5c, Supplementary Fig. 15).

Following changes were made on to figure 5 in the manuscript.

Changes in the Supplementary Information:

Following changes were made on to figure 15 in the SI.

Procedure for preparing samples for Circular Dichroism: 2 mg of wild type myoglobin and modified myoglobin were dissolved in 1 mL of water. 50 μ L of each were used to perform Circular Dichroism analyses. Circular Dichroism was recorded on a Jasco-810 Spectropolarimeter. Samples were micro-pipetted onto a 50 μ L Hellma Analytics quartz cell with a 0.1 mm path length (Model # 106-0.10-40). Spectra were measured by averaging three scans from 260-190 nm with a 0.2 nm data pitch and 100 nm s⁻¹ scanning speed.

5. A major challenge on targeting Lysine is the formation of heterogeneous conjugates, as encountered by the authors. Have any attempts on targeting the cysteine residues, using an external amine, been performed? Since the data points towards the thiol additions occurring before the condensation with amines, this approach should be viable and yield more homogeneous constructs.

We would like to thank the reviewer for pointing this out. Similar questions were asked by reviewer #1 too. We have conducted Cys and Lys duo labeling experiments using our chemistry (new data in Fig 6b) and carried out labeling of cysteine in proteins as analyzed by gels (new data added in figure 7b) to further demonstrate one of the potential applications of our method. Adjustment was made in both the manuscript and the SI. For the detail, please refer to answers for Q3 and Q5 of reviewer #1.

Reviewers' Comments:

Reviewer #1:

Remarks to the Author:

The authors have addressed most of the comments satisfactorily. It is interesting to see how the conformational flexibility of a peptide dominates its restrictive capabilities while driving selectivity between competing amines. This trend is likely to revert with a protein as a substrate. The authors can discuss their viewpoints in the manuscript. The experiments to gain selectivity at the cost of conversions are also impressive. Besides, I appreciate that the authors highlight the site-heterogeneity for single-site labeled proteins. While responding to reviewer 2, the authors highlighted NHS ester chemistry for comparison. However, the reviewer apparently points out the contributions from Hamachi, Francis, Rai, Bernardes, and Gois. The authors should consider placing a brief argument to put their work in this perspective.

Overall, this work is an excellent contribution to the chemical technologies for the precision engineering of proteins and should be considered for publication in Nature Communications.

Reviewer #2:

Remarks to the Author:

This reviewer is not satisfied with the revisions the authors made. As mentioned in the previous comments, this work showcased a synthetic application of a WELL-KNOWN 3-component reaction for peptide and protein modification. It is publishable, but publication in top-notch journals like Nat Commun is a big stretch in its current form. Several studies have reported the protein cross-linking induced by furan-derived cis-2-butene-1,4-dial (e.g. Bioorganic Chemistry 125 (2022) 105852, <https://doi.org/10.1016/j.bioorg.2022.105852>). A recent study by Zheng in ACS Chem Biol (2022, 17, 873, <https://doi.org/10.1021/acscchembio.1c00917>) demonstrated an elegant application of this reaction in profiling furan-protein adducts in cultured cells. With that said, the reviewer might still consider supporting its publication in this journal if the authors can rewrite the paper more candidly. It is unnecessary to rebrand the known reaction as FuTine. The previous contributions must be clearly and fairly acknowledged in both the main text and figures. The drawback of the new application should be clearly presented in the figures as well.

Reviewer #3:

Remarks to the Author:

The authors have improved the manuscript significantly by clarifying the questions raised adding new text, providing additional references and/or adding additional data. This new multicomponent reaction between a furan, a thiol, and an amine to yield modified pyrroles and its application in chemical biology warrants publication of this manuscript in Nature Communications in its current form.

Thank you for your consideration of this work. We thank the reviewers for critical reading of the manuscript. We hope you will consider a revised version that addresses the minor concerns of each reviewer.

We thank all the reviewers for the recommendation to publish after minor changes.

Our responses to the reviewers' comments are shown below.

Reviewer #1 (Remarks to the Author):

The authors have addressed most of the comments satisfactorily. It is interesting to see how the conformational flexibility of a peptide dominates its restrictive capabilities while driving selectivity between competing amines. This trend is likely to revert with a protein as a substrate. **The authors can discuss their viewpoints in the manuscript.** The experiments to gain selectivity at the cost of conversions are also impressive. Besides, I appreciate that the authors highlight the site-heterogeneity for single-site labeled proteins. While responding to reviewer 2, the authors highlighted NHS ester chemistry for comparison. **However, the reviewer apparently points out the contributions from Hamachi, Francis, Rai, Bernardes, and Gois.** The authors should consider placing a brief argument to put their work in this perspective. Overall, this work is an excellent contribution to the chemical technologies for the precision engineering of proteins and should be considered for publication in Nature Communications.

We thank the reviewer for the encouraging comments and willingness to accept it.

We have added the following lines in the revised manuscript:

1. "We do not expect to see similar observation with proteins where different lysines have different microenvironments influencing their pKa. Thus there would be a preference for a particular lysine over the others."

2. We have added the following references from Hamachi, Francis, Rai, Bernardes, and Gois and the following lines in a discussion related to the homogeneous labeling of proteins in the revised manuscript.

"Previous methods for labeling lysine either generated heterogeneous mixture of products modifying multiple lysine residues³²⁻³³ or form homogeneous products mediated by heterobifunctional probes.³⁴⁻³⁶ In the later cases, the length of linkers between two different functional groups dictated the regioselectivity, rather than the inherent reactivity of the lysine residues on a protein.³⁵⁻³⁶ Such methods have not been reported for chemoproteomic profiling of lysine."

References:

32. Cal, P. M. S. D. et al. Iminoboronates: A new strategy for reversible protein modification. *J. Am. Chem. Soc.* **134**, 10299-10305 (2012).

33. McFarland, J. M. & Francis, M. B. Reductive alkylation of proteins using iridium catalyzed transfer hydrogenation. *J. Am. Chem. Soc.* **127**, 13490-13491 (2005).
34. Matos, M. J. et al. Chemo- and regioselective lysine modification on native proteins. *J. Am. Chem. Soc.* **140**, 4004-4017 (2018).
35. Reddy, N. C. et al. Traceless cysteine-linchpin enables precision engineering of lysine in native proteins. *Nat. Commun.* **13**, 6038-6052 (2022).
36. Tsukiji, S. et al. Ligand-directed tosyl chemistry for protein labeling *in vivo*. *Nat. Chem. Biol.* **5**, 341–343 (2009).

Reviewer #2 (Remarks to the Author):

This reviewer is not satisfied with the revisions the authors made. As mentioned in the previous comments, this work showcased a synthetic application of a WELL-KNOWN 3-component reaction for peptide and protein modification. It is publishable, but publication in top-notch journals like Nat Commun is a big stretch in its current form. Several studies have reported the protein cross-linking induced by furan-derived cis-2-butene-1,4-dial (e.g. *Bioorganic Chemistry* 125 (2022) 105852, <https://doi.org/10.1016/j.bioorg.2022.105852>). A recent study by Zheng in *ACS Chem Biol* (2022, 17, 873, <https://doi.org/10.1021/acscchembio.1c00917>) demonstrated an elegant application of this reaction in profiling furan-protein adducts in cultured cells. With that said, the reviewer might still consider supporting its publication in this journal if the authors can rewrite the paper more candidly.

- We thank the reviewer willing to accept it after minor changes in writing. We agree that this work is an application for the well-known reaction and we have acknowledged it clearly throughout the manuscript. Similar to “click” chemistry a well-known reaction discovered by Sharpless in **1998** but later applied by Bertozzi in the year **2006** for varying biological applications.
- We demonstrate the application of FuTine MCR for the selective modification of peptides, and synthesis of macrocyclic and stapled peptides. This reaction was successfully applied for the modification of twelve different proteins of varying sizes and 3D structures with high efficiency. The resulting pyrrole-protein conjugates showed high stability in varying acidic and basic pH conditions. In addition, FuTine chemistry is compatible with other conjugation chemistry as shown by the post-modification of tyrosine residue by diazo coupling. We demonstrate the application of FuTine chemistry to generate homogeneous protein conjugates and its ability to distinguish subtle reactivity differences among lysines on native proteins is remarkable for protein engineering. We utilized FuTine chemistry for the selective labeling of lysine and cysteine on proteins with fluorophores in a complex cell lysate mixture. The reaction was utilized for the dual labeling of both lysine and cysteine on a protein in a selective manner by controlling the amounts of amine and thiol reagents. The reaction was deployed successfully for the homogeneous stapling of a protein between two lysine residues. Our discovery of harnessing FuTine chemistry for the precise homogeneous protein engineering, homogeneous stapling of proteins and chemoproteomic profiling in a complex cell lysate is a significant advance in protein modification as acknowledged by Reviewers #1 and #3 as well.

- We also did a thorough characterization of all the products by NMR or MS/MS and carried out chemo selectivity studies to confirm that the reaction is orthogonal between furan, thiol and amine, which is not reported in either of the above papers. **The major focus of all the previous papers is to identify cross-linked proteins in cells to identify the biomarkers and mechanisms behind furan carcinogenicity.**
- We also wanted to point out that the paper “Bioorganic Chemistry 125 (2022) 105852, <https://doi.org/10.1016/j.bioorg.2022.105852>.” showed that two different kinds of products are possible, 2-thio *N*-pyrrole and 3-thio *N*-pyrrole and the characterization of both the compounds was done by MS only. We also observed similar results with our peptide experiments, which we have confirmed by NMR of both the 2-thio *N*-pyrrole (**2h'** and **2f'**) and 3-thio *N*-pyrrole (**2h** and **2f**) products. Our results clearly showed that 2-thio *N*-pyrroles (**2h'** and **2f'**) were formed in much lower yields (5.3-14.7%) as compared to 3-thio *N*-pyrroles (**2h** and **2f**, 52.3-53.9%). Moreover, we showed that the formation of 2-thio *N*-pyrrole is dependent on amine substrates and the formation of 2-thio *N*-pyrrole was observed only with 2-aminoethanol and glycine (Fig. 3a, **2h'** and **2f'**) out of 12 examples with varying amines and thiols.

It is unnecessary to rebrand the known reaction as FuTine. The previous contributions must be clearly and fairly acknowledged in both the main text and figures.

As suggested, we added the previous contribution in **Fig. 1**. We have acknowledged the previous contribution throughout the manuscript in the main text. Please note the below lines that are already present in the manuscript and some new lines and references have also been added in the revised manuscript.

Inspired by the well-known cytochrome P450-catalyzed oxidation of furan to *cis*-2-butene-1,4-dial (BDA), which reacts with glutathione (GSH) and cellular amines to generate thio-*N*-pyrroles cross-linked products,¹⁹⁻²³ we re-envisioned this observation as a highly selective, multicomponent reaction (3-MCR) (Fig. 1).

New additions:

1. **Fig. 1. Bioinspired one-pot furan-thiol-amine (FuTine) multicomponent reaction (MCR).** Previous work: The identification of the toxicity of furan by analyzing the protein crosslinking products obtained by the reaction of amine and thiols on proteins with electrophilic *cis*-2-butene-1,4-dial (BDA) generated by oxidation of furan by enzyme P450.
2. “Previously this reaction has been extensively explored for determining the biomarkers associated with furan toxicity by analyzing the cross-linked proteins due to the low volatility of the furan.¹⁹⁻²³ Recently, Zheng et al explored this method for chemoproteomic profiling of lysine and cysteine on proteins in a complex cellular mixture.²⁴”
3. “A similar observation for the formation of minor 2-thio *N*-pyrrole has also been reported previously.²⁷”

4. “Inspired by the enzyme catalyzed toxicity of furan, which generates cross-linked proteins inside cells, we developed a robust and highly efficient one-pot multicomponent chemoselective reaction for coupling thiols and amines with furans to generate highly stable *N*-pyrrole products.”
5. “We demonstrate the application of FuTine MCR for the selective modification of peptides, and synthesis of macrocyclic and stapled peptides. This reaction was successfully applied for the modification of twelve different proteins of varying sizes and 3D structures with high efficiency. The resulting pyrrole-protein conjugates showed high stability in varying acidic and basic pH conditions. In addition, FuTine chemistry is compatible with other conjugation chemistry as shown by the post-modification of tyrosine residue by diazo coupling. We demonstrate the application of FuTine chemistry to generate homogeneous protein conjugates and its ability to distinguish subtle reactivity differences among lysines on native proteins is remarkable for protein engineering. We utilized FuTine chemistry for the selective labeling of lysine and cysteine on proteins with fluorophores in a complex cell lysate mixture. The reaction was utilized for the dual labeling of both lysine and cysteine on a protein in a selective manner by controlling the amounts of amine and thiol reagents. The reaction was deployed successfully for the homogeneous stapling of a protein between two lysine residues. Our discovery of harnessing FuTine chemistry for the precise homogeneous protein engineering, homogeneous stapling of proteins and chemoproteomic profiling in a complex cell lysate is a significant advance in protein modification. The majority of lysine modification approaches either typically generate a heterogeneous mixture of products modifying multiple lysine residues or a few generate homogeneous products typically requiring the heterobifunctional molecules joined by linkers.”

References:

21. Peterson, L. A. Reactive metabolites in the biotransformation of molecules containing a furan ring. *Chem. Res. Toxicol.* **26**, 6–25 (2013).
22. Peterson, L. A., Cummings, M. E., Vu, C. C. & Matter, B. A. Glutathione trapping to measure microsomal oxidation of furan to cis- 2-butene-1,4-dial. *Drug Metab. Dispos.* **33**, 1453-1458 (2005).
23. Gates, L. A., Lu, D. & Peterson, L. A. Trapping of cis-2-butene-1,4-dial to measure furan metabolism in human liver microsomes by cytochrome P450 enzymes. *Drug Metab. Dispos.* **40**, 596–601 (2012).
27. Muńko, M., Ciesielska, K. & Pluskota-Karwatka, D. New insight into the molecular mechanism of protein cross-linking induced by cis-2-butene-1,4-dial, the metabolite of furan: Formation of 2-substituted pyrrole cross-links involving the cysteine and lysine residues, *Bioorg. Chem.* **125**, 105852-105859 (2022).

The drawback of the new application should be clearly presented in the figures as well.

As suggested, we have added the following drawbacks of our methodology in the discussion in the revised manuscript.

“One of the drawbacks of the FuTine method for the modification of peptide and proteins is the low stability of BDA so it needs to be prepared fresh every time. Secondly, both

furan and thiol are sensitive to oxidizing conditions therefore BDA needs to be generated separately followed by the sequential addition of thiol and amine for obtaining a high yield of the conjugate.”

Reviewer #3 (Remarks to the Author):

The authors have improved the manuscript significantly by clarifying the questions raised adding new text, providing additional references and/or adding additional data. This new multicomponent reaction between a furan, a thiol, and an amine to yield modified pyrroles and its application in chemical biology warrants publication of this manuscript in Nature Communications in its current form.

We thank the reviewer for the encouraging comments and the willingness to accept it without any further changes.